

# Composite analysis of the tropopause inversion layer in extratropical baroclinic waves

Thorsten Kaluza[1], Daniel Kunkel[1], and Peter Hoor[1]

[1]Institute for Atmospheric Physics, Johannes-Gutenberg University Mainz, Mainz, Germany

**Correspondence:** Thorsten Kaluza (kaluzat@uni-mainz.de)

**Abstract.** The variability and similarities in the evolution of the tropopause inversion (TIL) layer during cyclongenesis in the North Atlantic storm track are investigated using operational meteorological analysis data (Integrated Forecast System from the European Centre for Medium-Range Weather Forecasts). For this a total amount of 130 cyclones have been analysed which evolved during the months August through October between 2010-2014 over the North Atlantic. Their paths of migration along
with associated flow features in the upper troposphere / lower stratosphere (UTLS) have been tracked using the mean sea level pressure. Subsets of the 130 cyclones have been used for composite analysis using minimum sea level pressure to filter the cyclones based on their strength.

The composite structure of the TIL strength distribution in connection with the overall UTLS flow strongly resembles the structure of the individual cyclones. Key results are that a strong dipole in tropopause inversion layer strength forms in regions
of cyclonic wrap-up of UTLS air masses of different origin and isentropic potential vorticity. These air masses are associated with the cyclonic rotation of the underlaying cyclones. The maximum values of enhanced static stability above the tropopause occur north and northeast of the cyclone centre, vertically aligned with outflow regions of strong updraft and cloud formation up to the tropopause, which are situated in anticyclonic flow patterns in the upper troposphere. These regions are colocated with a maximum of vertical shear of the horizontal wind. The strong wind shear within the TIL results in a local minimum of
Richardson numbers, representing the possibility for turbulent instability and potential mixing (or air mass exchange) within regions of enhanced static stability in the lowermost stratosphere.

## 1   Introduction

The tropopause inversion layer (TIL) is a ubiquitous feature of the upper troposphere/lower stratosphere (UTLS) region in equatorial, midlatitude, and polar regions (e.g., Birner et al., 2002; Gettelman and Wang, 2015). It is commonly defined as
a vertically confined layer of enhanced static stability and is usually analysed using the squared Brunt Väisälä frequency, $N^2 = g\Theta^{-1}\partial_z\Theta$ (Birner et al., 2002). In the extratropics, the TIL is co-located to a region of strong trace gas gradients between the troposphere and the stratosphere (Hegglin et al., 2009; Kunz et al., 2009; Schmidt et al., 2010), which define the extratropical transition layer (Ex-TL, Pan et al. (2004); Hegglin et al. (2009)), or mixing layer (Hoor et al., 2002, 2004). This co-location sometimes led to the assumption that the TIL might inhibit cross-tropopause transport (Hegglin et al., 2009; Get-
telman and Wang, 2015), however, evidence for this relation is still missing. Moreover, the TIL is essential for the vertical




propagation of waves on different scales, ranging from small scale gravity waves to large scale Rossby waves (e.g., Birner, 2006; Sjoberg and Birner, 2014; Gisinger et al., 2017). The sharp jump in static stability at the tropopause from mean tropospheric values of $N^2 = 1 \times 10^{-4}\,\mathrm{s}^{-2}$ to mean stratospheric values of $N^2 = 4 \times 10^{-4}\,\mathrm{s}^{-2}$ or to the even larger values defining the TIL results in a maximum of the so called refractive index controlling the upward propagation of waves, and leading to partial or even total wave reflection at the tropopause.

This study focusses on the evolution of the TIL at midlatitudes, where the flow in the UTLS is largely dominated by baroclinic planetary and synoptic scale waves. The role of such waves on the formation and maintenance of the TIL in midlatitudes has been the subject of a variety of scientific studies. Idealised modelling studies showed that the TIL can be formed due to conservative dynamics. Wirth (2003, 2004) performed potential vorticity (PV) inversions on axisymmetric PV anomalies of different sign in an idealised background atmosphere, pointing out an adiabatic sharpening mechanism of the lower stratospheric temperature profile related to the convergence of the secondary circulation vertical wind in anticyclonic flow. They were furthermore able to show that the advection of enhanced static stability from low to high latitudes plays an important role for the lower stratospheric $N^2$ maximum in anticyclonic flow. Wirth and Szabo (2007) performed baroclinic life cycle simulations with a comprehensive numerical weather prediction model and were able to confirm the concept of an adiabatic sharpening mechanism of the tropopause. Following up on these results, Erler and Wirth (2011) performed adiabatic baroclinic life cycle simulations with the same setup and concluded that breaking of baroclinic waves is an important process for the irreversible and permanent formation of a residual TIL as evident in the zonal or temporal mean state. Kunkel et al. (2014) performed similar baroclinic life cycle experiments with the focus on the impact of inertia-gravity waves on the thermal structure in the UTLS. They found that these waves, after being emitted from imbalances along the jet, modulate the ambient thermodynamic variables such as the static stability $N^2$ and persistently modify the TIL structure through the dissipation of the gravity waves. The role of diabatic processes in the TIL formation during baroclinic life cycle simulations was then studied by Kunkel et al. (2016), who attributed the relative to the adiabatic case stronger TIL evolution to diabatic processes related to moist dynamics and radiative effects of clouds reaching up to the tropopause. The stratospheric residual circulation also contributes significantly to the sharpening of the tropopause (Birner, 2010) especially at midlatitudes and during winter, where the downwelling in the extratropics induces a warming which lowers the tropopause and results in a strong localised positive forcing on the static stability.

Randel et al. (2007) performed radiative transfer model calculations to compare the radiative effect of realistic measurement-based mean ozone- and water vapour profiles to profiles with varying gradients of both constituents at tropopause height. They linked the strong gradients of ozone and water vapour at the tropopause to a dipole of the radiative forcing with cooling below and heating above the local tropopause. In turn this leads to an enhancement of static stability in the lower stratosphere. Pilch Kedzierski et al. (2015) analysed the synoptic scale behaviour of the TIL based on Global Positioning System (GPS-RO) radio occultation temperature profiles in combination with data from the European Centre for Medium-Range Weather Forecasts (ECMWF). They found the strongest TIL values in the midlatitudes within ridges and during winter, and thus confirmed and expanded previous more theoretical studies concerning the correlation between the TIL strength and the relative vorticity of the upper tropospheric flow. Pilch Kedzierski et al. (2017) applied a wavenumber-frequency domain filtering method on



GPS-RO temperature profiles, and were able to attribute a major part of the instantaneous TIL signal in midlatitudes to the transient and reversible modulations caused by planetary- and synoptic-scale waves. In conclusion, these previous works show that planetary and synoptic scale waves in the UTLS region play a major role on one hand concerning the instantaneous and potentially reversible sharpening of the lower stratospheric temperature gradients, as well as on the other hand the formation

of an irreversible and permanent residual background TIL.

The goal of this study is to complement the previous studies by analysing common structures of the TIL evolution in baroclinic waves over the North Atlantic. For this we use ECMWF operational analysis data over a five year period, and first focus on the evolution of the TIL in individual life cycles, and second derive composites of life cycles to analyse common patterns in the evolution of the lower stratospheric static stability over a set of 130 individual baroclinic life cycles over the North Atlantic.

The evaluation of average atmospheric properties with composites especially in the vicinity of cyclones was used in a variety of previous studies, and based on a variety of underlying data. Wang and Rogers (2001) analysed dynamical and thermal characteristics of explosive cyclones during a 12 year period over the North Atlantic, based on ECMWF analysis data. Catto et al. (2010) compared composites of the 100 most intense extratropical cyclones in the northern hemisphere from the 40-year ECMWF reanalysis (ERA-40) data set and the high resolution global environment coupled climate model (HiGEM), to assess

the capability of climate models to produce coherent airstream features, i.e. the warm conveyor belt, the cold conveyor belt and the dry intrusion. Recently Flaounas et al. (2015) studied a set of 200 intensive Mediterranean cyclones based on a 20 year Weather Research and Forecasting (WRF) regional model data simulation, with one focus among others on the UTLS PV forcing on the overall life cycle evolution and its synergy with the tropospheric development of the cyclones. To our knowledge the presented study is the first to focus on the TIL and correlated features in the context of cyclone composites.

The paper is structured as follows. In Sect. 2 we present the data set, the surface cyclone tracking algorithm and our approach to derive composites of different dynamical and thermodynamical variables in the UTLS. In Sect. 3 we illustrate the evolution of the UTLS features for two different life cycles which remarkably resemble the well known life cycles LC1 and LC2 from Thorncroft et al. (1993). In Sect. 4 we present composites of a variety of variables from different subsets of the cyclones emphasising the evolution of the TIL in baroclinic life cycles as well as associated flow features. We close our discussion in Sect.

5 by summarising our findings and putting them into perspective of previous studies.

## 2   Data and methods

### 2.1   ECMWF operational analysis data

For the detection of cyclone tracks we use operational analysis fields from the integrated forecast system (IFS) from the European Centre for Medium-Range Weather Forecasts (ECMWF), for August to October from 2010 until 2014. The spatial

extent of the area covers the North Atlantic from 60° W to 20° E and from 20° N to 75° N, and therefore encompasses the autumn maximum of Atlantic storm tracks (Wernli and Schwierz, 2006). We use six hourly available analysis fields during the given time period, with a grid spacing of 0.25°in the horizontal, a vertical resolution of L91 for the years 2010 until 2012, and L137 for 2013 and 2014. Moreover, the choice of the time period as well as the region of the data was motivated by the



preparation of the airborne measurement campaign WISE (Wave driven isentropic exchange) which took place over the North Atlantic in autumn 2017.

We decided to use the operational analysis data over e.g. the a more consistent reanalysis data set like ERA-Interim, due to the finer vertical resolution in the tropopause region. While ERA-Interim with 60 model levels has a vertical grid spacing of about 1 km in the UTLS, the operational analysis has a vertical grid spacing of about 300-400 m, depending on the tropopause location and on the vertical grid spacing of L91 and of L137. In particular, this leads to a much better representation of the static stability in the lower stratosphere. The formation of the TIL in numerical models is known to be sensitive to the horizontal and vertical resolution as well as their ratio (e.g., Birner, 2006; Wirth and Szabo, 2007; Son and Polvani, 2007; Erler and Wirth, 2011).

We use analysis data on model levels which provides the best vertical resolution in the UTLS of roughly 300 m. Many of the desired variables such as the temperature $T$, the three-dimensional wind $(u, v, \omega)$, the cloud ice water content $ciwc$ and relative vorticity $\zeta_{rel}$ are directly provided by the ECMWF, while other quantities have to be derived from the primary fields, such as static stability $N^2$, potential vorticity $PV$, vertical wind shear $S^2$, and the Richardson number $Ri$.

We define the strength of the TIL as the maximum in static stability within 3 km above the lapse rate tropopause. The lapse rate tropopause is defined as the lowest level where the temperature lapse rate falls below $2.0\,\mathrm{K\,km^{-1}}$ and its average between this level and all higher levels within 2 km above this level remains below this value (WMO, 1957). We do this, since the high resolution data shows large variability in the UTLS region, with often several maxima evident above the tropopause. Therefore, we find this definition of the TIL strength to be preferable over e.g. the first maximum in static stability above a threshold ($4 \times 10^{-4}\,\mathrm{s^{-2}}$, e.g., Gettelman and Wang, 2015).

## 2.2 Cyclone tracking

A major goal of this study is to analyse the evolution of dynamical features in the UTLS in life cycles of baroclinic waves, and link these with the evolution of the static stability $N^2$ above the tropopause. Baroclinic life cycles up to the point of breaking are often associated with surface cyclones (e.g., Thorncroft et al., 1993), and the flow in the UTLS above these cyclones is an important region in regard to the enhancement in static stability above the tropopause. Several methods are available to trace cyclones, using e.g. the associated maximum in relative vorticity on lower oder middle tropospheric pressure levels, or the minimum in mean sea level (MSL) pressure. The IMILAST experiment (Neu et al., 2013) showed that many of these methods achieve comparable results. We tested several methods with short time periods and with comparable results. Ultimately, we decided to use the sea level pressure field due to the smoothness of this field compared to e.g. the relative vorticity, which makes it easier to identify cyclone centres. Our algorithm therefore identifies surface cyclones in the MSL pressure field and traces them in time and space.

The tracking algorithm is based on Hanley and Caballero (2012), and searches local minima in the MSL pressure field. Since our data has a fine horizontal grid spacing and is limited to the North Atlantic, we had to partly adapt the tracking algorithm to our data set. The major steps are 1.) smoothing of the MSL pressure field, 2.) identification of all local minima at all time steps, and 3.) the connection of the local minima from consecutive time steps to cyclone tracks. The following paragraph will



give more details.

A local minimum in a gridded MSL pressure field is defined as a grid point having a lower value than its surrounding 8 grid points. To reduce the amount of local MSL pressure minima found at each time step, a Cressman filter (Cressman, 1959) is applied, averaging each grid point in the field with its neighbouring grid points within a radius $r < r_0$ ($r_0$ being 500 km), using weights of $(r_0^2 - r^2)/(r_0^2 + r^2)$. The smoothed field exhibits less local minima, reducing the amount of criteria needed to define a cyclone center, without altering the tracks of the cyclones fundamentally. After applying the Cressman filter and following once more Hanley and Caballero (2012), the MSL pressure field at each time step is projected onto an area-preserving Lambert projection centred at the North Pole, to counteract the bias in zonal resolution caused by the convergence of the meridians on the native latitude-longitude-grid. The projected MSL pressure field is then interpolated onto a regular equidistant grid with 28 km grid spacing, which corresponds to the $0.25°$ horizontal resolution at the equator. The algorithm now searches and saves every local minimum in the MSL pressure field, with two extra criteria being applied: 1.) an upper threshold of $p_t = 1007.25$ hPa, and 2.) the neglection of all minima located over orography higher than 1500 m. The first criterium replaces the pressure gradient criterium applied in Hanley and Caballero (2012), since the limitation to a regional domain makes it difficult to calculate consistent pressure gradients. The value for $p_t$ was determined by testing several values below 1013.25 hPa, with 1007.25 hPa being the largest value of minimum MSL pressure where our algorithm was able to connect the local minima to coherent cyclone tracks. Our algorithm therefore neglects very weak minima in the pressure field, but since weak cyclones or cyclones in very early/very late stages of their life cycles are often not strongly connected to the upper tropospheric flow, it is sufficient for this study to track them not from their very first nor until their very last appearance. Also we are focussing on the time periods around the mature stage of the baroclinic waves, i.e. when the MSL pressure reaches its lowest values. The second criterion is another result of the IMILAST experiment (Neu et al., 2013) and is applied due to the error associated with reducing the surface level pressure to sea level from such altitudes.

In the next step the algorithm connects minima from consecutive time steps by searching in a given radius for the nearest minimum. For minima associated with a new formed cyclone the search radius in the second time step is 720 km from the position where the cyclone first appears. For minima already existing for two or more time steps the algorithm follows Wernli and Schwierz (2006) with a 'first guess' approach, where the first guess location of the cyclone is a linear continuation of the track in latitude-longitude-coordinates: $\mathbf{x}^*(t_{n+1}) = \mathbf{x}(t_n) + 0.75[\mathbf{x}(t_n) - \mathbf{x}(t_{n-1})]$. Wernli and Schwierz (2006) introduce the factor of 0.75 because cyclone movement tends to get slower during a cyclone's life cycle. The corresponding MSL pressure minimum is then defined as the nearest minimum from $\mathbf{x}^*(t_{n+1})$ within a radius of 840 km. For more information concerning the values of the search radii and the first guess approach see Hanley and Caballero (2012) and Wernli and Schwierz (2006). Following yet another result from the IMILAST experiment (Neu et al., 2013) only cyclones with a lifetime of at least 24 hours are further considered which translates to at least five 6-hourly time steps in the IFS analysis data. The algorithm furthermore neglects cyclones with less than two time steps before and/or after the global minimum in MSL pressure along their path to make sure that the actual intensification period is covered by the data. We want to emphasize that due to these two criteria only extratropical cyclones are selected. Tropical cyclones in extratropical transition emerging from the western edge of our data region which might have a strong signal in the MSL field but no real intensification period are neglected by the algorithm.





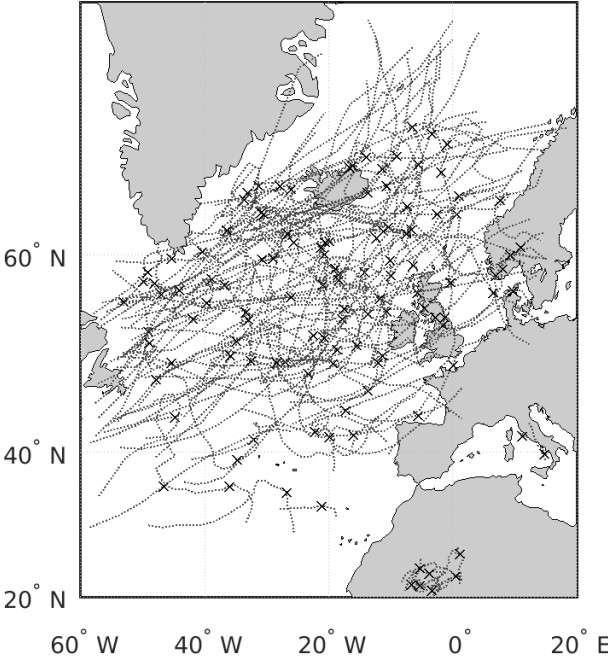

**Figure 1.** Mercator projection of the area covered, with all 139 cyclone tracks found by the algorithm. Crosses indicate the location of minimum sea level pressure along each track.

Figure 1 shows the cyclone paths tracked by the algorithm after applying all criteria. The distribution of tracks matches well with the climatological cyclone frequencies described in Wernli and Schwierz (2006). Aside from the large accumulation over the North Atlantic there are also several tracks located over North Africa and the Mediterranean Sea. The relatively small number of tracks over the Mediterranean Sea can be explained by the $p_t = 1007.25$ hPa upper limit criterium, and the fact that

5     these Mediterranean cyclones hardly exhibit strong minima in the MSL pressure field. This study focusses on Atlantic storm tracks, therefore the cyclones over North Africa an the Mediterranean Sea are sorted out by a geographical criterium.

### 2.2.1 Composites of extratropical cyclones

Ultimately, we want to analyse the variability of the tropopause inversion layer within extratropical baroclinic waves. For this, we compute composites of the cyclones at the time of maximum intensity which we define as the occurrence of the global

10    minimum surface pressure along the track. We select a subset of the gridded data for each cyclone by rotating the pole of a spheric polar coordinate system $(\Theta, \phi)$ onto the centre of the cyclone. The horizontal resolution of the new coordinate system is set to 0.25° and the radius to $\Theta_{max} = 15°$ (Figure 2). The radius is chosen such that all relevant features around the cyclone centre are covered. The rotation of the new coordinate system onto the cyclone centre is performed following Bengtsson et al. (2007), who provide a detailed description of the rotation matrix in their appendix. The original ECMWF IFS variables



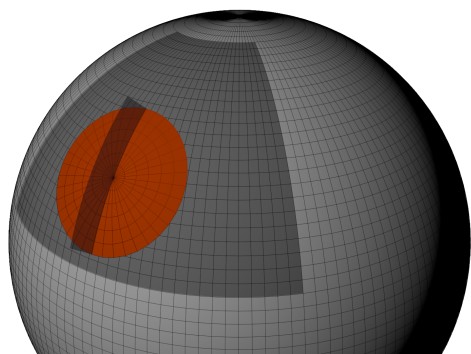

**Figure 2.** Schematic of the size of the analysed region (dark grey) in comparison to the full sphere, as well as the size of a rotated spheric polar coordinate system (orange) with $\Theta_{max} = 15°$ radius. The dark grey semi-transperent upright plane illustrates (with exaggerated vertical extent) the vertical crosssection aligned from north to south as displayed in Figure 9.

in latitude-longitude coordinates are then interpolated onto a pillar covered by the new coordinate system, with a lapse rate tropopause based vertical coordinate and a vertical grid spacing of $\Delta z = 100 \ m$. The interpolation onto the new coordinate system based on the centre of the surface cyclone does not account for a potential vertical tilt of the cyclone. However since this study focusses on the point in time of maximum cyclone intensity when the vertical structure tends to align, we find a tilt

to be negligible. The ensemble of tropopause based pillars of variables from each cyclone is then averaged to create a three-dimensional composite of the flow in the vicinity of the cyclones. Subsequently, the absolute height coordinate is recovered by calculating the mean tropopause height at each horizontal location.

In the special case of composites of horizontal or quasi-horizontal variables like the potential vorticity on an isentropic surface or the TIL strength, we first calculate the fields for each cyclone and then afterwards the mean. This method preserves more

information because the three-dimensional tropopause based averaging still smoothes vertical information due to the variability of e.g. the height of the maximum in $N^2$ above the tropopause. In contrast to other studies analysing cyclone composites (e.g., Bengtsson et al., 2007; Catto et al., 2010), we keep the orientation of each individual cyclone instead of rotating them dependent on their path of migration. In our case this approach leads to a better representation of the dynamical and thermodynamical features in the UTLS.

# 3   The lower stratospheric static stability evolution during two baroclinic wave breaking events

Before we present the result of the cyclone composite, we first discuss two cases of individual cyclones and associated TIL evolution over the North Atlantic. We choose these cases since both cyclones are associated with upper tropospheric baroclinic wave breaking events very similar to the ones described in idealised baroclinic life cycle simulations. The first case study shows an evolution comparable to an LC2 (life cycle 2), while the second one exhibits distinct features of an LC1 (life cycle 1) wave



breaking event (Thorncroft et al., 1993). Although the waves occur consecutively in time and space, they are not interacting directly and the cyclones at surface level evolve relatively isolated from each other, in contrast to multi-cyclone-centres as described for example by Hanley and Caballero (2012).

5 **3.1 Baroclinic wave with LC2 characteristics**

Figure 3 shows three consecutive time steps of the evolution of the first baroclinic wave with the time of maximum intensity depicted in the second row. We focus our analysis and discussion on the flow features inside the 15° radius around the cyclone centre. Furthermore, we focus our discussion on sea level pressure, isentropic potential vorticity (IPV) at 330 K, strength of the tropopause inversion layer as defined in the beginning of Sect. 2, and relative vorticity at the lapse rate tropopause (from

10 left to right in Figure 3).

24 hours before the cyclone reaches maximum intensity the mean sea level pressure already falls below 975 hPa. The IPV shows a baroclinic wave centred above the tracked surface cyclone centre, with large IPV values coming from high latitudes and tilting in northwest-southeast direction into the jet. During the intensification of the surface cyclone the upper air wave enters what Thorncroft et al. (1993) call the cyclonic wrap-up phase which in this case study exhibits distinct features of an

LC2. During the wrap-up which is vertically aligned with the strong surface cyclone the trough stays on the northern cyclonic shear side of the jet streak as indicated by the 200 hPa horizontal wind maximum in Figure 3d. The wave breaking event is meridionally confined by the jet streak and only a very thin streamer of enhanced IPV air turns anticyclonically on the last day depicted in Figure 3. The relative vorticity at lapse rate tropopause height correlates with the IPV where cyclonic (anticyclonic) flow exhibits large (low) values of IPV. The regions with relatively small (large) values of IPV and anticyclonic (cyclonic) flow

exhibit enhanced (reduced) values of static stability above the tropopause. This correlation especially holds true inside the 15° radius area and at the point in time of maximum cyclone intensity. This agrees well with the anticipated role of balanced dynamics (Wirth, 2003, 2004) and idealised baroclinic life cycle simulations with focus on the evolution of the TIL (e.g., Erler and Wirth, 2011). The regions with the largest values of static stability up to $N^2 = 10 \times 10^{-4} \, \text{s}^{-2}$ are located inside the ridge in the first time step and later in the second time step in the tropospheric IPV air mass which is wrapped up around the

underlaying cyclone centre. In the last time step the TIL strength especially within the 15° radius reverts to lower values. We want to highlight the fact that the enhancement of the TIL strength during the wave breaking event is not uniform, but rather shows a large spatial variability and wave-like patterns on different horizontal scales.

The TIL strength at lower latitudes in the quasi-horizontal illustration (Figure 3c) sometimes exhibits strong gradients or

jumps which appear non-coherent. These strong gradients are sort of an evaluation artefact, due to 1.) the restriction of the TIL strength to the global maximum in static stability within 3 km above the lapse rate tropopause, 2.) the high variability of the tropopause defining lapse rate in the high resolution data, 3.) the occurrence of sharp tropopause jumps and double tropopauses (especially during wave breaking events), and 4.) the strongly structured TIL itself. The occurence of such features should be kept in mind when analysing such synoptic situations in a high resolution data set. We compared such prominent structures





**Figure 3.** Evolution of a baroclinic wave breaking event as seen in ECMWF IFS analysis data. The middle row represents the 14th of October 2014 at 06:00 UTC, the point in time of maximum cyclone intensity, the upper and lower row show the situation 24 hours prior and past the maximum intensity. Column a.) shows the pressure at mean seal level (solid lines $p_{msl} < 1013$ hPa, dashed $p_{msl} > 1013$ hPa, dotted $p_{msl} = 1013$ hPa), b.) the IPV on 330 K, c.) $N^2_{max}$ as indicator for the TIL strength, and d.) the relative vorticity at lapse rate tropopause height. Blue lines in middle row show 40 and 50 ms$^{-1}$ horizontal windspeed isolines at 200 hPa. Dashed black line shows the path of migration of the tracked cyclone, with the position of the cyclone centre at the point in time of the meteorological field displayed marked by the black x. Black circles show the $15°$ radius used for the composites. Note the deformation of the circles due to the mercator projection.





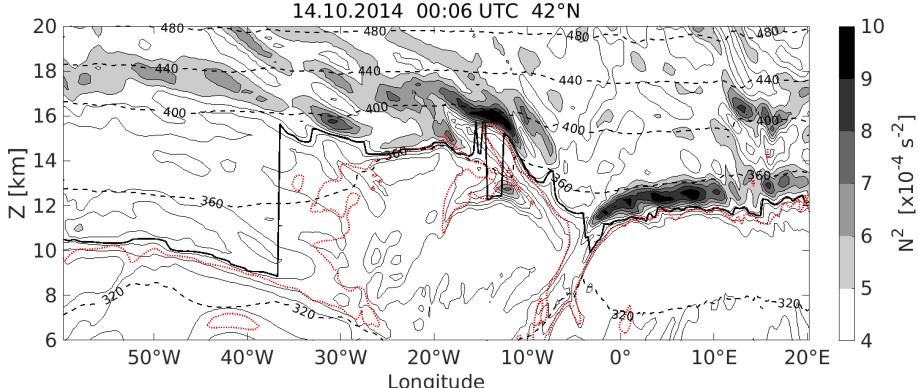

**Figure 4.** Vertical crosssection over the North Atlantic on 14.10.2014 at 00:06 UTC, and at $42°$ N. Filled contour as well as thin solid black contour lines show static stability $N^2$ in steps of $1 \times 10^{-4} s^{-2}$, dashed black lines isentropes. The bold solid black line indicates the lapse rate tropopause, the dotted red line the 2 pvu isoline of potential vorticity.

in the quasi horizontal illustration of the TIL with a variety of vertical crosssections, to ensure that the statements we make about the TIL evolution hold true and are in fact not a numerical evaluation artefact. Figure 4 shows an example for a vertical crosssection corresponding to the second time step depicted in Figure 3 and at $42°$ N. It illustrates the large variability of the three dependent variables PV, static stability $N^2$ and the lapse rate tropopause height. The jump of the lapse rate tropopause at

about $37°$ W as well as the small scale stratospheric PV intrusion at about $12°$ W illustrate the complexity of the tropopause location and the challenge to account for this by the different definitions of the tropopause and the TIL.

### 3.2 Baroclinic wave with LC1 characteristics

Figure 5 shows the subsequent baroclinic wave breaking event 3 days later over the North Atlantic. The mean sea level pressure on 16 October still shows two distinct minima, the decaying northern one associated with the previous wave breaking event,

as well as a newly formed minimum. The background state of the UTLS is still significantly distorted due to wave breaking event of the LC2 described in Sect. 3.1. Similar to the previous case a relatively small scale baroclinic wave is evident in the IPV field above the underlaying surface cyclone centre. In contrast to the previous case, as the surface cyclone grows stronger and the upper air wave enters the wrap-up phase (Figure 5 middle row), the initially cyclonically tilted trough with enhanced values of IPV turns anticyclonically and later on a substantial part of the trough penetrates the jet in its excursion southwards.

The jet splits into two jet streaks (middle row Figure 5d.) and the trough gets thinned, eventually producing a cut-off (Figure 6). These wave breaking characteristics meet the definition of an LC1 as described by Thorncroft et al. (1993). During the evolution of the cyclone the relations between tropospheric IPV, anticyclonic relative vorticity at tropopause height, and an enhancement in static stability above the tropopause are all evident. The regions of maximum static stability with values up to $N^2 = 10 \times 10^{-4}$ s$^{-2}$ are again located inside the ridge of low IPV air wrapping up around the underlaying cyclone centre





**Figure 5.** As in Figure 3, but for the subsequent cyclone with maximum intensity on 17 October 2014 at 18:00 UTC, depicted in the middle row. Top row 24 hours earlier respectively bottom row 24 hours later.



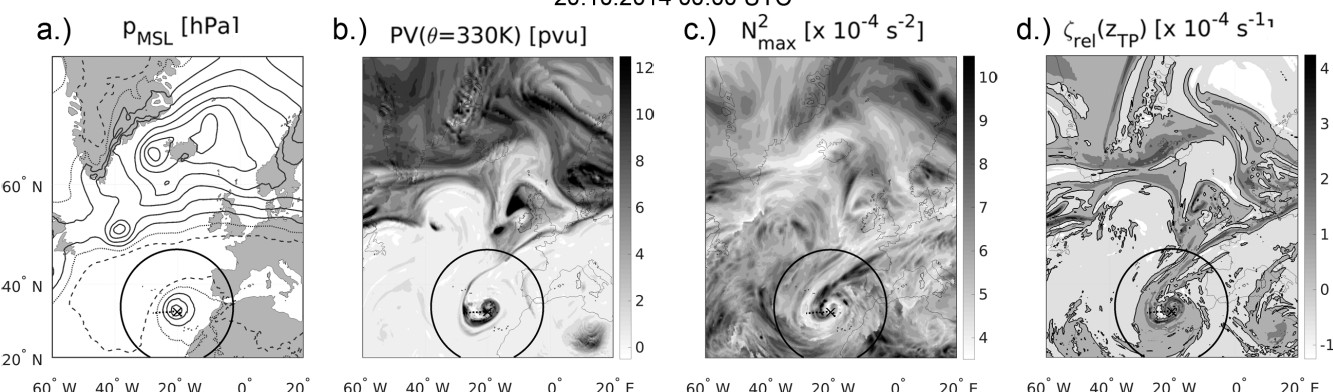

**Figure 6.** As in Figure 3, but only for one point in time, showing the tracked cyclone associated with the upper tropospheric cut-off from the previous baroclinic wave breaking.

about the time of maximum cyclone intensity. The thinning and southward moving trough itself exhibits low values of static stability $N^2$ above the tropopause, in agreement with its positive relative vorticity, but the flow around the trough shows no distinct strong signal in the quasi-horizontal static stability distribution, especially when compared to the cyclonic wrap-up 24 hours earlier.

Figure 6 shows a comparable evolution about two days later for the secondary cyclone associated with the cut-off which formed from the thinning streamer. This cyclone was also tracked by the algorithm and while it is weaker and exhibits less horizontal extend than the two previous cases, the region of maximum static stability with values of up to $N^2 = 10 \times 10^{-4} \mathrm{s}^{-2}$ still evolves inside the wrapped up low-IPV air. The wrapped up cut-off exhibits maximum static stability values of about $N^2 = 4 \times 10^{-4} \mathrm{s}^{-2}$. The regions of high static stability above the tropopause are horizontally coherent with the occlusion as

well as the region of outflow of ascending air masses into the jet.

     In conclusion, we analysed two subsequent baroclinic life cycles, the first resembling an LC2, and the second an LC1 when compared with idealised baroclinic life cycle simulations. In both cases the regions of strongest enhancement in static stability above the tropopause are located inside the ridge of low-IPV air moving northward from low latitudes and wrapping up around the underlaying primary cyclone about the time of maximum cyclone intensity. The flow around the southward excursion of

the trough in the LC1 wave exhibits no distinct signal, until a secondary cyclone associated with the cut-off from the trough evolves. This *cyclone-linked* behaviour can also be seen in the idealised baroclinic life cycle simulations from Erler and Wirth (2011). Their Figure 4 shows a comparable evolution of the TIL strength with consideration that there are two northern surface cyclone intensification periods and one cut-off related cyclone forming at low latitudes.

     Based on the analysis of these two case studies, we further motivate the analysis of the evolution of the TIL during baroclinic

life cycles based on the flow above surface cyclones. We recognise that surface cyclones exist which are not linked to baroclinic





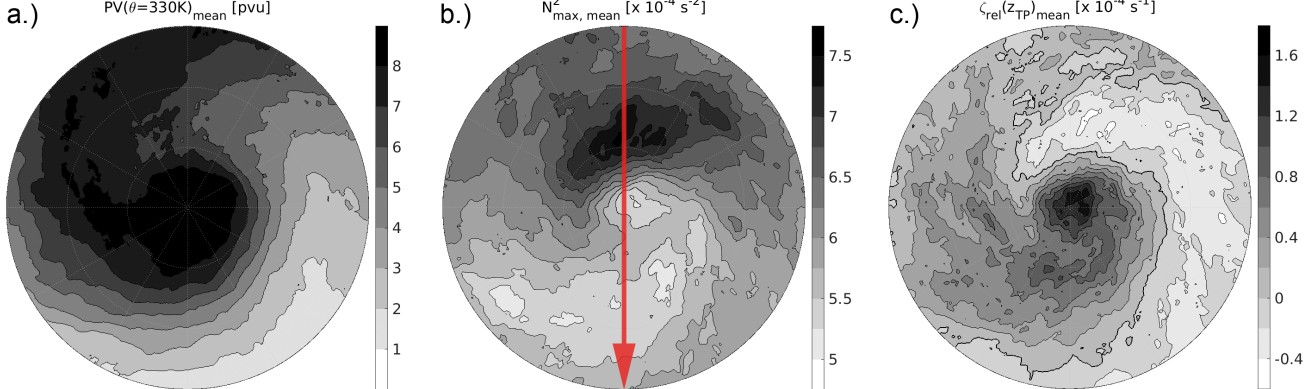

**Figure 7.** Composite of flow features for the 76 strong cyclones with $p_{msl,\,min} \leq 990$ hPa at the point in time of maximum intensity. From left to right, IPV on 330 K, TIL strength $N^2_{max}$, and relative vorticity $\zeta_{rel}$ at lapse rate tropopause height. The $\zeta_{rel} = 0\,\mathrm{s}^{-1}$ isoline is indicated by the black solid line. The red arrow in the middle indicates the orientation of the crosssection in Figure 9.

wave breaking events and also baroclinic wave breaking events (in the sense of meridional irreversible redistribution of isentropic potential vorticity) exist which are not linked to surface cyclones. Nevertheless, we expect a composite of strong surface cyclones to give a characteristic representation of the flow where a strong TIL forms during the subset of those baroclinic wave breaking events which are in fact linked to surface cyclones.

## 4    Composite analysis of extratropical cyclones over the North Atlantic

In the following section we present composites of upper tropospheric / lower stratospheric flow features to define a mean characteristic evolution of the flow features in the UTLS above surface cyclones during baroclinic wave breaking events. We compute composites of subsets of the tracked cyclones based on the surface cyclone strength, because we expect the strength to be a good indicator for the amount of coupling between the surface cyclone and the flow in the UTLS. We abstain from presenting composites from cyclone subsets based on other criteria because the amount of cyclones tracked in our data set quickly reduces to a statistically non-significant amount when specific criteria are applied.

Figure 7 shows the quasi-horizontal composites of selected features from the 76 strong cyclones with $p_{msl,\,min} \leq 990$ hPa. The three contour plots show a mean state of the UTLS flow which is in its basic features comparable to the ones discussed in the two case studies of baroclinic wave breaking events. The first contour plot shows a streamer of stratospheric IPV on 330 K reaching from north-west into the cyclone centre, with a cyclonic rotational component. Naturally, the lowest values of IPV are located in the south and gradually approach the IPV values of the stratospheric streamer when rotating counterclockwise around the cyclone centre along the wrap-up. This region of strong with respect to the cyclone centre tangential gradients of IPV exhibit to large parts a negative mean relative vorticity at tropopause height and also large values of static sta-





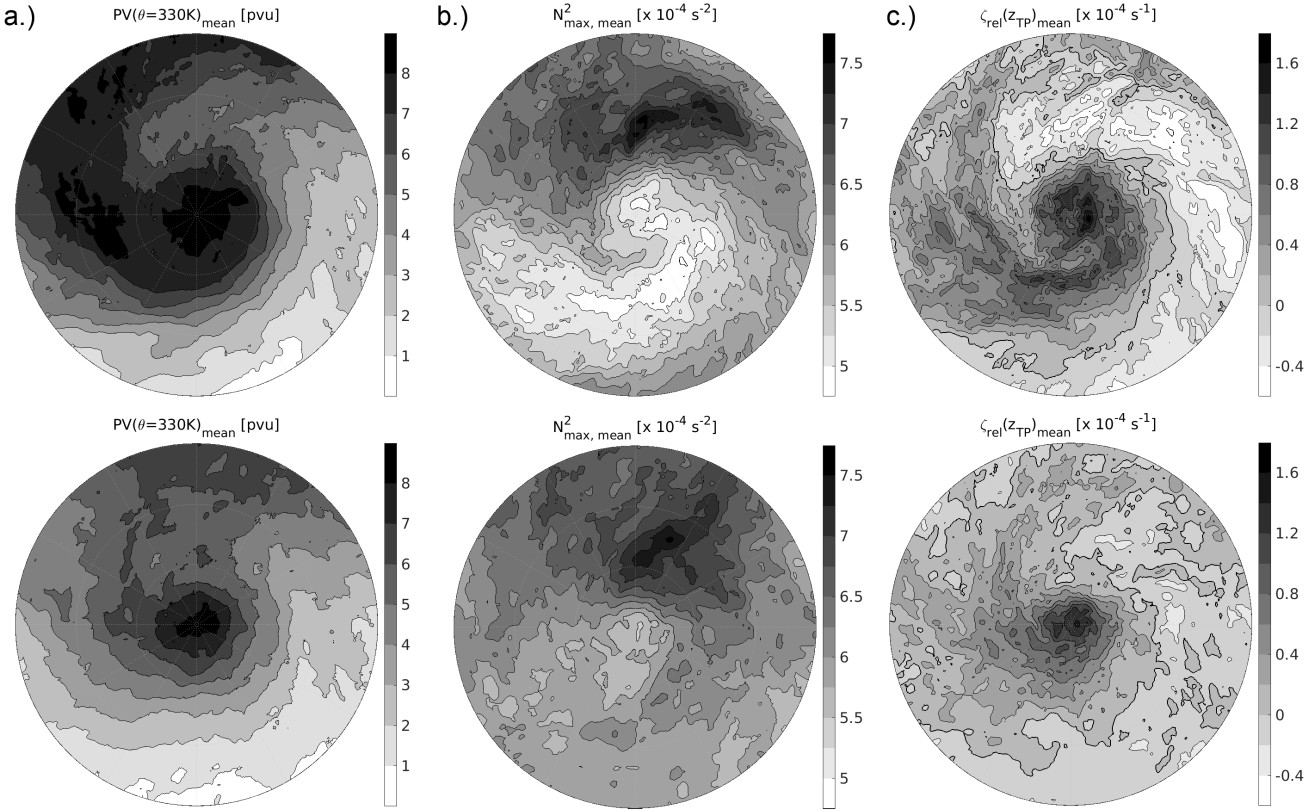

**Figure 8.** As in Figure 7, but for 1.) top row: for the 30 very strong cyclones with $p_{msl,\,min} \leq$ 980 hPa, and 2.) bottom row: the 54 weak cyclones with $p_{msl,\,min} >$ 990 hPa.

bility in the lower stratosphere. The maximum values of static stability are located north of the cyclone centre with maximum values up to $N^2_{max,\,mean} = 7.5 \times 10^{-4}$ s$^{-2}$. The stratospheric IPV streamer as well as the southern regions of strong radial IPV gradients exhibit a cyclonic relative vorticity at tropopause height as well as a relatively weak TIL with values of about $N^2_{max,\,mean} \approx (5 - 5.5) \times 10^{-4}$ s$^{-2}$.

In Figure 8 the same contour plots are displayed for a subset of very strong cyclones as well as a subset of weak cyclones. The composites in the top row represent the 30 very strong cyclones with $p_{msl,\,min} \leq$ 980 hPa. The dominating cyclonic rotational component is stronger pronounced and the air masses with distinctly different features are wrapped up around the cyclone centre to a larger degree. The relative vorticity and the TIL strength still correlate well and their horizontal gradients

10 are sharper compared to those in Figure 7. The latter is not caused by a stronger TIL in case of $p_{msl,\,min} \leq$ 980 hPa, but more due to a clearer separation and also a significant decrease in TIL strength in the cyclonic flow.





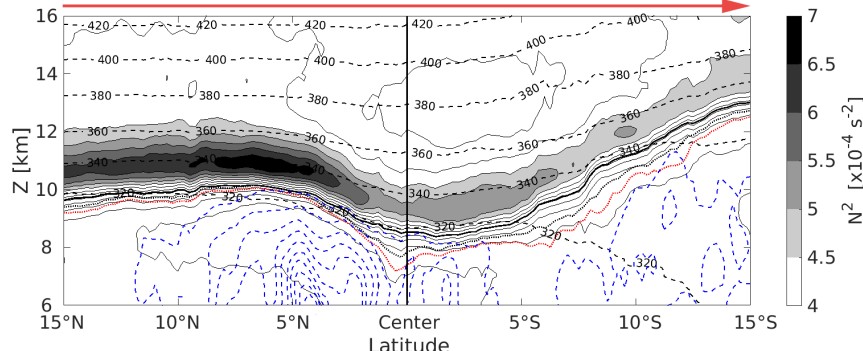

**Figure 9.** Vertical crosssection of the mean state from the 76 strong cyclones, from north to south through the centre of the cyclone, as indicated by the red arrow here and in Figure 6. The crosssection is composed of the tropopause based mean of vertical profiles at each latitude. The mean tropopause height has been restored. Filled contour as well as solid black contour lines show static stability $N^2$ in steps of $0.5 \times 10^{-4} \mathrm{s}^{-2}$, dashed black lines isentropes, and dashed blue lines the cloud ice water content (ciwc) with the first isoline at and steps of $5 \times 10^{-6}$ kg/kg. The bold solid black line indicates the lapse rate tropopause, the dotted red (black) line the 2 pvu (2.5 pvu) isoline of potential vorticity, indicating near-tropopause PV gradients.

The 54 weak cyclones with $p_{msl,\,min} >$ 990 hPa are presented in the bottom row of Figure 8. The features described for the subsets of the strong and the very strong cyclones are still evident, however, with smoother gradients in all three features depicted and a weaker separation between tropospheric and stratospheric (influenced) air masses. The region of maximum enhancement of static stability above the tropopause is still located north of the cyclone centre and also still reaches mean

values of about $N^2_{max,\,mean} \approx (7 - 7.5) \times 10^{-4}\ \mathrm{s}^{-2}$.

Figure 9 shows a mean vertical crosssection from north to south through the cyclone centre for the 76 strong cyclones with $p_{msl,\,min} \leq$ 990 hPa along the red line as indicated in Figure 7. The thermal and dynamical tropopauses have both minimum height in the cyclone centre and both slope upward south of the cyclone centre. Furthermore, the regions of maximum TIL

strength partly align vertically above regions of ascending air and cloud formation in the troposphere indicated by the cloud ice water reaching up to the lapse rate tropopause. This hints towards the role of moist dynamical processes in the formation of the TIL (Kunkel et al., 2016). To analyse the region of strongest enhancement in static stability north of the cyclone centre in more detail we derive the composites shown in Figure 10. Depicted are the vertical wind, the squared vertical shear of the horizontal wind $S^2$, and the gradient Richardson number similar to the northern half of the crosssection in Figure 9 and for

the same subset of cyclones ($p_{msl,\,min} \leq$ 990 hPa). The mean vertical wind $\omega$ correlates well with the mean cloud ice water content. The stratosphere shows no distinct mean vertical velocity signal and the troposphere far north from the cyclone centre exhibits slowly descending air masses. The squared vertical shear of the horizontal wind speed $S^2 = (\partial u/\partial z)^2 + (\partial v/\partial z)^2$





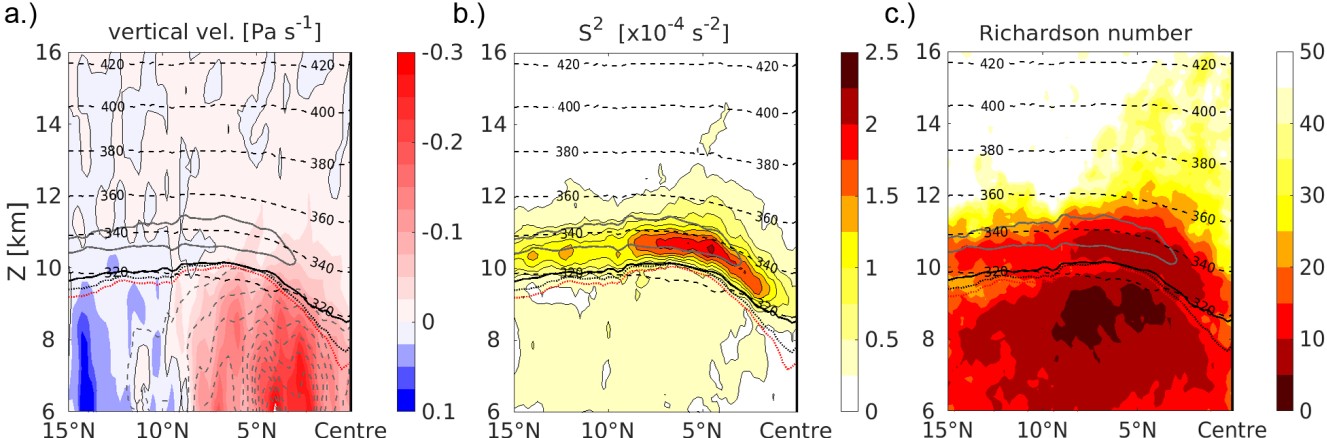

**Figure 10.** Vertical crosssection as in Figure 8, but only north the cyclone centre. Left panel: filled contours show mean vertical velocity, solid thin black line the $\omega = 0\,\mathrm{Pa\,s^{-1}}$ isoline. Grey dashed lines are isolines of the cloud ice water content, steps as in Figure 9. Middle panel: filled contour and solid black contour lines show squared vertical shear of horizontal wind $S^2$. Right panel: modified Richardson number mean $Ri$. Grey solid bold line in all three panels shows the $N^2_{mean} = 6 \times 10^{-4}\,\mathrm{s^{-2}}$ isoline, dashed black lines the isentropes.

shows a maximum above the tropopause, with a remarkable overlap into the regions of maximum static stability. We calculate the modified Richardson number mean as described in Birner et al. (2002) who take the natural logarithm of the individual Richardson numbers before computing an average to reduce the variability range: $Ri' = \mathrm{sign}(N^2)\ln(|N^2|/S^2 + 1)$. The mean Richardson number is then calculated as $Ri = \exp(\overline{Ri'}) - 1$. The result is depicted in Figure 10 and shows two regions of

minimum mean Richardson numbers. The lowest values of $Ri$ are evident in the upper troposphere in the upper regions of the ascending air masses which is a result of a very weak stratification combined with a moderately strong vertical wind shear due to a local maximum of the meridional wind component at tropopause height at about 7.5° N (not shown). A second region of minimum mean Richardson numbers is located right above the tropopause near the cyclone centre, and extends into the regions of maximum static stability. This agrees well with one result from Birner et al. (2002) who saw a similar partly vertical overlap

of a maximum in vertical wind shear and the TIL in radiosonde profiles. Grise et al. (2010) furthermore stated a fundamental link between the zonal-mean static stability and the zonal-mean zonal wind based on the thermal wind relation, highlighting a dependency between the meridional gradient of the static stability and the curvature of the vertical shear of the horizontal wind.

We furthermore tested different methods of calculating an average Richardson number to check the robustness of our re-

sult. Calculating the unmodified Richardson numbers for each individual cyclone and averaging afterwards produces a very fragmented mean field, but it still exhibits a region with local minima of the order of $10^1$ inside the region of the TIL. Calculating the mean Richardson number from the averaged potential temperature and wind field yields a result more comparable to the modified Richardson mean, but with overall larger values for $Ri$, stronger separated minima, and sharper gradients of

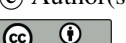


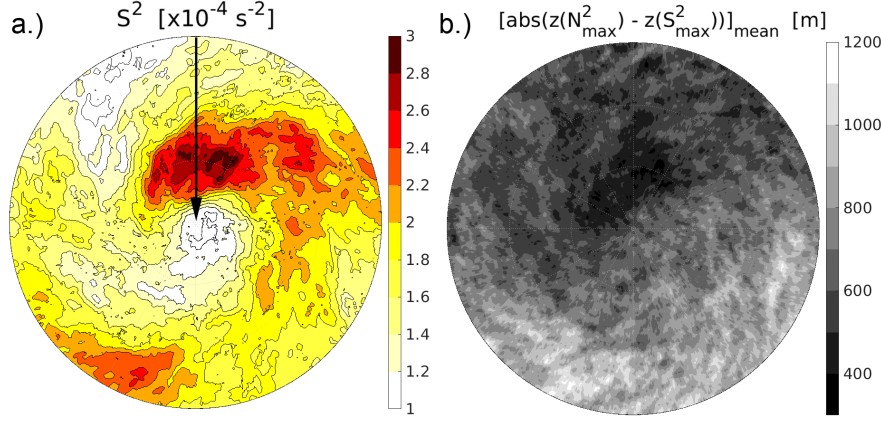

**Figure 11.** Composite of flow features for the 76 strong cyclones with $p_{msl,\,min} \leq 990$ hPa at the point in time of maximum intensity. Left panel: composite of the maximum in vertical wind shear $S^2_{max}$ found within 3 km above the lapse rate tropopause for each cyclone. The black arrow indicates the position of the vertical crosssections in Figure 9. Right panel: composite of the absolute height difference between the maximum in static stability $N^2_{max}$ and the maximum in vertical wind shear $S^2_{max}$, both within 3 km above the lapse rate tropopause.

$Ri$. Birner et al. (2002) observed a distinct discrepancy in the vertical shear of the horizontal wind $S^2$ above the tropopause between the radiosonde data and ECMWF reanalysis data. This shows that the vertical wind gradients in the UTLS are not well resolved and significantly underestimated by the reanalysis data, a tendency which might still be the case in the analysis data. The fact that we see a strong maximum in $S^2$ above the tropopause in NWP data, while Birner et al. (2002) saw none in

the mean profiles from reanalysis data, can be explained by 1.) the finer vertical and horizontal resolution in the operational IFS model, and 2.) the fact that we present a composite of a specific synoptic situation which may result in a similar structure of the individual vertical profiles. Richardson numbers of the order of 5-10 still represent a stable flow, but we want to stress that these are mean values from 76 cyclones, and based on vertical gradients of $N^2$ and $S^2$ derived from a vertical grid spacing of about $\Delta z \approx 300$ m. Therefore, there is the possibility that turbulence is present even in regions of enhanced static stability

in the lower stratosphere which might affect cross tropopause transport in these regions.

Figure 11 provides a strong indication that a colocation between the regions of enhanced static stability above the tropopause and the maximum in vertical shear of the horizontal wind exists. Figure 11 a.) shows the quasi-horizontal composite of the maximum in wind shear within 3 km above the tropopause for the subset of strong cyclones ($p_{msl,\,min} \leq 990$ hPa). It reveals together with Figure 7 that there is a colocation between the regions of maximum squared wind shear $S^2_{max}$ above the

tropopause and the regions of maximum enhancement in static stability $N^2_{max}$ in the lower stratosphere. Figure 11 furthermore shows how both features are also vertically constrained north of the cyclone centre, resulting in the vertical overlap of the wind shear region and the TIL (see also Figure 9 and Figure 10 b). Overall the composites give a comprehensive overview over the





mean evolution of the TIL above surface cyclones and its relation with other flow features like tropospheric moist diabatic active regions or wind shear in the lower stratosphere.

## 5    Discussion and conclusions

The goal of this study was to analyse the development of the TIL during the evolution of baroclinic life cycles in operational
ECMWF IFS analysis data. For this we tracked a total sum of 130 surface cyclones in the MSL pressure field over the North Atlantic from August to October during a five year period from 2010 to 2014. We presented an analysis of two consecutive individual cyclones associated with two upper tropospheric baroclinic wave breaking events, one resembling an LC1 and the other one an LC2 event. We furthermore derived composites of the atmospheric state from different subsets of the 130 cyclones using the intensity of the surface cyclone as a selection criterion.

Based on the analysis of the two individual baroclinic life cycles (Sect. 3) in comparison with idealised baroclinic life cycle simulations from previous works (Wirth and Szabo, 2007; Erler and Wirth, 2011; Kunkel et al., 2014, 2016), we identified the regions of cyclonic wrap-up around the cyclone centre as the regions exhibiting the strongest signal in static stability above the lapse rate tropopause. The flow in the UTLS with relatively large values of isentropic potential vorticity and positive relative vorticity coming from northwest and wrapping up around the cyclone centre exhibits low values of static stability, while the
counterpart of anticyclonic low-IPV air originating from lower latitudes shows a distinct enhancement in static stability and represents the region with a well developed TIL. The largest values of static stability above the tropopause are located north and northeast of the cyclone centre at about the time of maximum surface cyclone intensity. The synoptic scale TIL evolution in both the LC1 and LC2 baroclinic waves is similar, with differences in the strength and the shape of the region of the wrap-up. The relatively weak secondary cyclone associated with the cut-off which resulted from the LC1 wave breaking event shows a
similar evolution on a smaller horizontal scale. The TIL above the individual cyclones exhibits a large temporal, horizontal, and vertical variability on different scales associated to the variety of known forcing mechanisms being resolved in the high resolution NWP data.

Furthermore, we presented composites of the atmospheric state in the vicinity of different subsets of tracked cyclone centres at the point in time of maximum cyclone intensity. We find that stronger surface cyclones are associated with a sharper and more
pronounced wrap-up in the UTLS flow. The composites furthermore resemble to a large degree the key features of the TIL evolution and the mean flow as identified from the case studies. The regions of largest TIL enhancement are located north and northeast of the cyclone centre above the occlusion and above regions influenced by strong tropospheric updrafts and clouds reaching up to the tropopause. This indicates the importance of moist dynamical and radiative processes during the formation of the TIL (e.g., Randel et al., 2007; Kunkel et al., 2016). The composites further reveal a maximum in vertical shear of the
horizontal wind $S^2$ within the region of strongest enhancement of static stability above the tropopause. The regions of maximum static stability and those of maximum wind shear show a remarkable overlap, horizontally as well as vertically, which is in agreement with previous studies (Birner et al., 2002; Grise et al., 2010). Richardson numbers calculated for these flow conditions favourable for turbulence reveal a region of local minima in $Ri$ right above the tropopause at around $5°$ north from



the cyclone centre. This result points toward a co-location between an enhancement in static stability above the tropopause and potential turbulent mixing of tropospheric and stratospheric air masses (Kunkel et al., 2016).

We want to note that baroclinic life cycles vary in their appearance from case to case and thus the TIL evolution in individual cyclones can differ from the one described by the composite analysis. The analysis of baroclinic life cycles in other regions

and seasons would of course be desirable, but is left open at this stage for later studies. The approach used in this study is now applicable to a large data set, e.g. using the new ERA-5 reanalysis which has the same vertical resolution in the UTLS as the analysis data in this study.

Overall this study confirms the importance of baroclinic waves and their embedded cyclones to explain the meso-scale variability of enhanced static stability above the lapse rate tropopause in the extratropics. The high spatial and temporal resolution

of the analysis data gives a better understanding on where and when static stability increases during baroclinic life cycles. The composites of baroclinic life cycles show a large agreement of TIL evolution with the individual life cycles in the analysis data as well as with the individual life cycles from earlier idealised studies (Erler and Wirth, 2011; Kunkel et al., 2016). They furthermore indicate that turbulent mixing might occur in regions of enhanced static stability right above the tropopause.

*Author contributions.* DK, PH and TK designed the research project. TK developed the model code and performed the calculations, and

analysed the data with the help of DK and PH. TK prepared the manuscript with contributions from all authors.

*Competing interests.* The authors declare that they have no conflict of interest.

*Acknowledgements.* The Authors acknowledge funding from the German Science Foundation, as this study was carried out as part of the preparation phase for the WISE campaign under funding from the HALO SPP 1294 (DFG grant no. KU 3524/1-1, HO 4225/7-1 as well as HO 4225/8-1). We are furthermore grateful to the ECMWF for providing the IFS analysis data which has been downloaded from the

ECMWF MARS system.





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
