# Peer review of "Composite analysis of the tropopause inversion layer in extratropical baroclinic waves"

_Atmospheric Chemistry and Physics, 2018_

## Referee Comment (RC1) · Anonymous Referee #1 · 20 Dec 2018

The study analyses the influence of cyclones on the tropopause inversion layer. The work builds on previous studies of baroclinic waves and performs of a detailed analysis of cyclone composites using ECMWF analysis data to show the transient behaviour of the static stability maximum. The last part of the study also briefly looks at the possibility that the high vertical shear in the TIL could lead to turbulence. Whilst I think that the study is detailed and interesting, I have several comments that the authors should address before publication.

Comments:

- Cyclone tracking: P5, lines 6 to 11. I appreciated the detailed description of the cyclone tracking algorithm but I am concerned about the effect of the projection and interpolation. The final composited feature in static stability looks like a North-South

dipole. I am wondering what the effect of the original grid and interpolation does to this feature. E.g., does it enhance the dipole?

- Composites of cyclones: P6, line 11. Do you mean cylindrical rather than spheric? The data you have is on an equidistant grid (Is this gridding only done for the cyclone tracking or is this gridded data used for the subsequent composites). I am confused about which data is being interpolated to the red pillar since the underlying grey grid in figure 2 is not equidistant.

- LC1 and LC2 case studies: Seeing the case studies is helpful in interpreting the composites. Care should be taken in discussing the static stability strength, particularly with the discontinuities that are seen as a result of the analysis in fig 3c and 5c. I would be interested in seeing the average of $N^2$ (as opposed to $N^2$ max) in the region 3 km above the tropopause. Side note: Add your definition of $N^2$ max to the caption in Fig 3c.

- In figure 4, it may be more helpful to show the cross section at some latitude north of the cyclone centre since this is the region where there is an enhancement in $N^2$.

- Composite analysis: I have some concerns about the compositing the values of $N^2$ max and artefacts that might arise as a result of this. Have you looked at a number of cyclones in your composite to make sure that the features in $N^2$ are indeed present in most of them?

- Richardson number analysis: I find this very interesting. Do the corresponding plots for the case studies in section 3 show very low values of Ri above the tropopause (regions with Ri <1)?

-P18 line 31. The Richardson numbers found are not low enough for turbulent flow. I would suggest not making such a strong statement in the conclusion.

- The colour scale on some of the figures could be chosen to be slightly more intuitive. E.g., Fig 10 (c) At first glance, I thought the red values were bigger.

Other comments:

- P2, line 31. Fix reference - P4 line 3 "the a more", line 25 "oder" - P6 line 6, "North Africa an the" - P9 caption a) "seal level" - P12 line 7 "extend" - P13 lines 17 and 18 "This region of strong with..." Meaning unclear. - P14 line 8 "stronger" to more pronounced - P 16 fig 10 caption "Vertical crosssection as in figure 8" should be figure 9. also "north the cyclone", "of" missing. -

---

## Referee Comment (RC2) · Anonymous Referee #2 · 14 Jan 2019

This paper investigates the tropopause inversion layer (TIL) strength using the maximum Brunt Vaisala frequency within 3km above the tropopause in 2 individual extratropical cyclone lifecycles, and in composites of strong extratropical cyclones in the Northern Hemisphere. This is following on from a number of studies analysing the TIL in idealised model simulations of baroclinic lifecycles. In both the case studies and the composites the authors find that the TIL strength (i.e. the highest values of static stability above the tropopause) can be found in the region of low isentropic PV that is advected cyclonically around the cyclone at upper levels. This is above the location of the ascent in the cyclones, where the clouds are identified. These results seem to be consistent with the previous studies.

I have a few comments, queries and suggestions that the authors should consider.

[Figure]

1. I found the introduction rather hard to read, with a lot of jargon that would be impenetrable to anyone new to the topic. I recommend reworking a lot of the language in the introduction section to make it clearer. For example, the 2nd paragraph on page 2 could be reworded. 2. Also regarding the introduction, it could be made clearer exactly where the gaps lie that this paper seeks to address. At the bottom of page 1, "evidence for this relation still missing" implies that this paper will provide evidence, but I am unsure if this is a goal of the study. Some rewording might make the introduction clearer overall. 3. Page 2 line 19: The "residual TIL" is mentioned more than once. Is it possible to define what this looks like? 4. Page 2 line 32: This Kedzierski et al reference is never returned to in the discussion section, although I am sure the results from the present paper confirm those results (the TIL strongest within the ridges). 5. Page 3, line 33: L91 and L137 are not vertical resolutions, but the number of levels. Could you give more information about the actual vertical resolution here? 6. Page 6, line 16: Could you make clearer what is meant by a "lapse rate tropopause based vertical coordinate"? Also page 7, line 7: what is meant by "the absolute height coordinate is recovered by calculating the mean tropopause height at each horizontal location"? 7. Page 7, line 8: What is meant by "horizontal or quasi-horizontal variables"? Does this just mean the horizontal composites of particular variables? 8. Page 8, line 11: Please do not start sentences with numerals. 9. Figure 3: The dotted MSLP contour is hard to distinguish. 10. Page 8, line 30: Is the maximum $N^2$ above the tropopause the best measure of TIL strength? It does seem to correlate well with the PV pattern, but would an average value give similar but smoother results? Have you tested this? 11. Figure 4: The caption seems to have incomplete sentences. 12. Page 12, lines 9-10: I'm unsure of what this is referring to. Can you point out where the occlusion and the jet are located? 13. Page 12, lines 17-18: This sentence is unclear. Did those authors look at the same events? 14. Pages 15-16: With the discussion of the Richardson number, it would be useful to the reader to have an explanation of what the results really mean. On page 17, lines 7-9, it seems to imply that despite the low Ri, turbulent mixing may occur, whereas in the discussion section (p18, lines 32-33) it seems

to say that because of the low Ri turbulent mixing occurs. Please could you clarify this. In relation to this, are you suggesting that the strong TIL actually enhances the mixing across the tropopause? This would be in opposition to the Heggelin et al 2009 and Gettelman and Wang 2015 references on page 1 of the introduction. 15. Page 18, line 26: "strong tropospheric updrafts"... I wonder here if what you mean is the ascent associated with the warm conveyor belt (e.g. Madonna et al ), which is where you would expect to see the diabatic heating. In Figure 3 it looks like the strongest TIL is clearly associated with the WCB at each time in the lifecycle. After the maximum intensity, the WCB anticyclonic outflow region (to the northeast of the cyclone) is the location of the strongest TIL. I think you need to include reference the WCB and how the observed features relate to it and the associated outflow. Ref: Madonna, E., H. Wernli, H. Joos, and O. Martius (2014a), Warm conveyor belts in the ERA-Interim dataset (1979–2010). Part I: Climatology and potential vorticity evolution, J. Clim., 27, 3–26, doi:10.1175/JCLI-D-12-00720.1.

---

## Author Comment (AC1) · 26 Feb 2019

**General comment from the referee:**

*"The study analyses the influence of cyclones on the tropopause inversion layer. The work builds on previous studies of baroclinic waves and performs of a detailed analysis of cyclone composites using ECMWF analysis data to show the transient behaviour of the static stability maximum. The last part of the study also briefly looks at the possibility that the high vertical shear in the TIL could lead to turbulence. Whilst I think that the study is detailed and interesting, I have several comments that the authors should address before publication."*

**Authors response:**

We thank the reviewer for the careful reading and interest in the study which helped to improve the paper. We reply to every comment in the order as they are listed in the review. We will respond in the following structure: First each comment from the reviewer, followed by our response, and finally the changes in the manuscript. The text coloured in blue are the changes associated with this review, cyan coloured text refers to the second review. The pages and lines indicate the position of the changes in the updated manuscript (the version without color-marked changes).
* * *
**Comment 1:** Cyclone tracking: P5, lines 6 to 11. I appreciated the detailed description of the cyclone tracking algorithm but I am concerned about the effect of the projection and interpolation. The final composited feature in static stability looks like a North-South dipole. I am wondering what the effect of the original grid and interpolation does to this feature. E.g., does it enhance the dipole?

**Reply to comment 1:** The Lambert projection is solely part of the tracking algorithm. Once the tracks are defined, all data like e.g. the quasi-horizontal fields of the maximum in static stability $N^2$ above the thermal tropopause are extracted from the original ECMWF latitude-longitude-grid. Therefore the Lambert projection has no effect on the analysis of the North-South dipole feature of the static stability.

The interpolation onto an area-preserving Lambert projection is done based on the work of Hanley and Caballero (2012) who chose this specific projection to counteract the bias in effective zonal resolution when searching for local minima in the mean sea level pressure field (or any field for that reason). The higher zonal resolution at higher latitudes (with reference to the geometric distance between grid points) could be regarded as an advantage, representing denser information, but at the same time it represents an inconsistency between high and low latitudes. The interpolation on to the area preserving Lambert projection with uniform geometric grid (with 28 km grid spacing, representing a form of lowpass-filter for the higher latitudes) counteracts that fact, achieving a more consistent cyclone track density over the whole area.
* * *
**Comment 2:** Composites of cyclones: P6, line 11. Do you mean cylindrical rather than spheric? The data you have is on an equidistant grid (Is this gridding only done for the cyclone tracking or is this gridded data used for the subsequent composites). I am confused about which data is being interpolated to the red pillar since the underlying grey grid in figure 2 is not equidistant.

**Reply to comment 2:** The basis for the interpolation is defined by the rotation of a spherical polar coordinate surface onto the centre of a tracked cyclone. The pillar of gridded data interpolated on the new rotated grid with the height coordinate dz relative to the tropopause could indeed be described as a cylindrical polar coordinate system. The underlaying grey grid in Figure 2 represents the original latitude-longitude grid as provided by the ECMWF with equidistant latitude and longitude grid spacing, but the geometric distance depends on the latitude (the aforementioned bias in zonal resolution). These original fields are interpolated onto the rotated new spherical coordinates with spherical cap of $15°$ (red area). This achieves comparable areas independent of where each cyclone centre is located. We reworded the specific paragraph describing the interpolation method as follows:

**Changes in the manuscript:**

**Page 7 Line 4:** We select a subset of the gridded data as provided by the ECMWF for each cyclone by rotating the pole

of a spheric polar coordinate system $(\Theta, \phi)$ onto the centre of the cyclone and interpolated the original data onto that new grid. The horizontal resolution of the new coordinate system is set to $0.25°$ and the spherical cap radius to $\Theta_{max} = 15°$ . The radius is chosen such that all relevant features around the cyclone centre are covered. Figure 2 illustrates the interpolation of the data from the original latitude-longitude grid provided by the ECMWF (light grey area for the whole sphere, dark grey for the limited area used in this study) onto an arbitrarily placed new spherical polar coordinate system (orange area).
* * *
**Comment 3:** LC1 and LC2 case studies: Seeing the case studies is helpful in interpreting the composites. Care should be taken in discussing the static stability strength, particularly with the discontinuities that are seen as a result of the analysis in fig 3c and 5c. I would be interested in seeing the average of $N^2$ (as opposed to $N^2_{max}$) in the region 3 km above the tropopause. Side note: Add your definition of $N^2_{max}$ to the caption in Fig 3c.

**Reply to comment 3:** We agree that a detailed discussion of the sub-synoptic-scale or even sub-meso scale features should be treated with caution when using a certain TIL strength definition, due to the discontinuities described. We nevertheless choose the maximum in $N^2$ above the tropopause as the TIL-strength definition for this study, because we are working with fine grid spacing data, and we want to highlight the variability of the static stability on different scales on one hand and show the difficulties when working with a certain definition of the TIL-strength on the other hand. The average within a certain vertical extent above the tropopause does not only filter a lot of information by default, but in particular for the feature of $N^2$ above the tropopause, because we know from previous work that larger maxima in $N^2$ above the tropopause tend to have a smaller vertical extent. Therefore an average over a defined vertical range above the tropopause could have a particular smoothing effect concerning even the large scale variability. We plotted the average of $N^2$ within 3 km above the tropopause for our case studies for comparison (Figure R1 and R2). Importantly, the analysis using a 3 km-averaged static stability shows very similar locations of the maximum TIL-structures on a synoptic scale, with the largest values in the regions of cyclonic wrap-up around the cyclone centre of air masses originating from lower latitudes. We did not add these plots to the manuscript, but we added the description of an average-based TIL strength definition at the end of section 2.1, to clarify that we deliberately decided to use another definition. We furthermore added the definition of the TIL strength to the caption in Figure 3.

**Changes in the manuscript:**
**Page 4 Line 18:** We  use this specific TIL strength definition, since the high resolution data shows large variability in the UTLS region, with often several maxima evident above the tropopause. Therefore, we find this definition of the TIL strength to be preferable over e.g. the first maximum in static stability above a threshold ($4 \times 10^{-4}\,\mathrm{s}^{-2}$, e.g., Gettelman and Wang, 2015). Another way to describe the static stability above the tropopause is to calculate the mean value of $N^2$ over a predefined vertical extent relative to the tropopause (e.g., Kunkel et al., 2014). While such a TIL strength definition achieves very similar results on a synoptic scale to the ones we will describe in Sect. 3, it also filters a lot of small scale variability and therefore partly neutralises the advantage of analysing high resolution data.

**Page 9 Fig. 3 caption:** Evolution of a baroclinic wave breaking event as seen in ECMWF IFS analysis data. The middle row represents the 14th of October 2014 at 06:00 UTC (the point in time of maximum cyclone intensity), the upper and lower row show the situation 24 hours prior and past the maximum intensity. Column (a) shows the pressure at mean  sea level (solid lines $p_{msl} \leq 1013$ hPa, dashed lines $p_{msl} > 1013$ hPa, in steps of 5 hPa), (b) the IPV on 330 K, (c) the maximum of static static stability $N^2_{max}$ within 3 km above the thermal tropopause as indicator for the TIL strength, and (d) the relative vorticity at lapse rate tropopause height. Blue lines in middle row show 40 and $50\,\mathrm{m\,s}^{-1}$ horizontal windspeed isolines at 200 hPa. Dashed black line shows the path of migration of the tracked cyclone, with the position of the cyclone centre at the point in time of the meteorological field displayed marked by the black x. Black circles show the $15°$ radius used for the composites. Note the deformation of the circles due to the mercator projection.

[Figure]

**Figure R1.** Evolution of the $N^2$ average based TIL strength for the LC2 resembling baroclinic wave breaking event. The grey contour shows the average of static stability over the vertical extent of 3 km above the thermal tropopause. Figure 1a) shows the TIL strength 24 hours before maximum intensity of the surface cyclone, b) at maximum intensity and c) 24 hours later.

[Figure]

**Figure R2.** As in Figure R1 but for the LC1 resembling baroclinic wave breaking event.

**Comment 4:** In figure 4, it may be more helpful to show the cross section at some latitude north of the cyclone centre since this is the region where there is an enhancement in $N^2$.

**Reply to comment 4:** We chose this cross-section in particular because it shows the small-scale stratospheric intrusion at around $12°$ W which is also visible as a discontinuity in the $N^2_{max}$ contour plot in Figure 3 in the middle row. It furthermore covers the tropopause jump between $30°$ W and $40°$ W. Both characteristics are the ones described in the paragraph associated with Figure 4. We included an additional cross-section (Figure R3) north of the cyclone centre into the manuscript to furthermore illustrate the variability of $N^2$ above the tropopause in the region of interest.

**Changes in the manuscript:**
Page 10 Line 25: The vertical cross-section in Fig. 4b shows the TIL structures at $60°$ N on the same day and north of the cyclone centre. It illustrates that the regions of enhanced static stability exhibit less variability at this latitude. They are located above the ridge between $40°$ W and $10°$ W, with the wave-like horizontal pattern that is also visible in Fig. 3c.

[Figure]

**Figure R3.** Vertical crosssection over the North Atlantic on 14.10.2014 at 06:00 UTC at $60°$ N. The filled contour as well as the thin solid black contour lines show static stability $N^2$ in steps of $1 \times 10^{-4}\mathrm{s}^{-2}$, dashed black lines show isentropes. The bold solid black line indicates the lapse rate tropopause, the dotted red line shows the 2 pvu isoline of potential vorticity.

**Comment 5:** Composite analysis: I have some concerns about the compositing of the values of $N^2_{max}$ and artefacts that might arise as a result of this. Have you looked at a number of cyclones in your composite to make sure that the features in N?2 are indeed present in most of them?

**Reply to comment 5:** This is a very valid concern which we shared during the data evaluation, especially since we know about the sensitivity of the result on the TIL-strength definition. We did in fact look at a large number of cyclones independently. While there is a large variability in the evolution of the upper tropospheric / lower stratospheric wrap-up around the cyclone centre (caused by the variability of the background flow, the cyclone evolution and the wave activity in the UTLS, as well as the variety of mechanism which influence the static stability at tropopause altitude). Most of the cyclones and especially the strong ones exhibit a comparable evolution of the static stability above the tropopause, with the wrapped-up North-South-dipole as the synoptic to mesoscale shaping. We furthermore estimate the mentioned discontinuities in the TIL-strength due the reasons described in the last paragraph of subsection 3.1 to be small and infrequent enough to be negligible for the large scale composite analysis. We updated Figure 11 and added Fig 12 to the manuscript with a set of vertical tropopause-based

cross-section composites of $S^2$ and $N^2$, which further illustrate the issue in a more comprehensive way. The plots complement Figure 9 and Figure 10b for different rotation angles around the cyclone centre, and while all these cross-section composites are solely based on the lapse rate tropopause definition, they do agree very well with the composites derived from the quasi-horizontal fields of the individual cyclones that depend on the definition of the TIL-strength and the strength of the wind shear (maximum of $N^2$ (Fig. 7b) / $S^2$ (Fig. 12) within 3 km above the tropopause). We added a description of this comparison to the manuscript.

[revised manuscript text omitted]

**Comment 6:** Richardson number analysis: I find this very interesting. Do the corresponding plots for the case studies in section 3 show very low values of $Ri$ above the tropopause (regions with $Ri < 1$)?

**Reply to comment 6:** The regions of enhanced static stability within the individual cyclones are mostly dominated by large vertical shear of the horizontal wind which results in low Richardson numbers despite the enhanced $N^2$, as is the case for the LC1 and LC2 case study (Figure R4). Several cyclones in our dataset exhibit regions with even very low Richardson numbers ($Ri < 1$) at tropopause altitudes, in anticyclonic upper tropospheric flow inside the ridge of the baroclinic waves and with partial overlap into the regions of enhanced static stability. These regions however appear as transient and localised features. A detailed analysis of such a case is subject to a corresponding study we are working on, which is in preparation to be submitted to ACPD.

[Figure]

**Figure R4.** Minimum Richardson numbers $Ri_{min}$ within 3 km above the thermal tropopause, at the point in time of maximum cyclone intensity a.) for the LC2 resembling baroclinic wave, and b.) for the LC1 resembling baroclinic wave.
* * *
**Comment 7:** P18 line 31. The Richardson numbers found are not low enough for turbulent flow. I would suggest not making such a strong statement in the conclusion.

**Reply to comment 7:** We rephrased the paragraph to make the statement less strong and to clarify that we do not expect the whole region to be dynamically unstable (because the Richardson numbers are still too large as you pointed out). We rather want to note that the general tendency towards high vertical wind shear in the region of interest (considering the variability of $N^2$ and $S^2$) points toward the possibility for localised turbulence.

**Changes in the manuscript:**
**Page 21 Line 11:** Mean Richardson numbers calculated for these flow conditions  reveal a region of local minima in $Ri$ right above the tropopause at around 5° north from the cyclone centre in the composite.  Taking into account the variability of $N^2$ and $S^2$ for individual cyclones, this result points toward the possibility of a co-location between enhanced static stability above the tropopause and

localised regions of turbulent mixing of tropospheric and stratospheric air masses (Kunkel et al., 2016).

**Comment 8:** The colour scale on some of the figures could be chosen to be slightly more intuitive. E.g., Fig 10 (c) At first glance, I thought the red values were bigger.

**Reply to comment 8:** We agree that this plot can be counterintuitive due to the inverted color scale, but we want to highlight the region with the tendency toward turbulent motion (i.e. low Richardson numbers) and therefore want to keep the inverted colors for this particular Figure. We added a note on this to the Figure caption. We furthermore changed the color scale to a lower maximum value to make the plot more easy to read.

**Changes in the manuscript:**
**Page 18 Fig. 10 caption:** Vertical tropopause-based averaged cross-section as in Fig. 9, but only north of the cyclone centre. Left panel: filled contours show mean vertical velocity, solid thin black line the $\omega = 0 \ \mathrm{Pa \, s^{-1}}$ isoline. Grey dashed lines are isolines of the cloud ice water content, steps as in Fig. 9. Middle panel: filled contour and solid black contour lines show squared vertical shear of horizontal wind $S^2$. Right panel: modified Richardson number mean $Ri$, with an inverted color scale (red represents low values) to highlight the tendency toward turbulent motion. Blue solid bold line in all three panels shows the $N^2_{mean} = 6 \times 10^{-4} \ \mathrm{s^{-2}}$ isoline, dashed black lines the isentropes.

**Other comments:** P2, line 31. Fix reference - P4 line 3 "the a more", line 25 "oder" - P6 line 6, "North Africa an the" - P9 caption a) "seal level" - P12 line 7 "extend" - P13 lines 17 and 18 "This region of strong with..." Meaning unclear. - P14 line 8 "stronger" to more pronounced - P 16 fig 10 caption "Vertical crosssection as in figure 8" should be figure 9. also "north the cyclone", "of" missing.

**Reply:** We fixed the mistakes in the manuscript.

---

## Author Comment (AC2) · 26 Feb 2019

**General comment from the referee:**

*"This paper investigates the tropopause inversion layer (TIL) strength using the maximum Brunt Vaisala frequency within 3km above the tropopause in 2 individual extratropical cyclone lifecycles, and in composites of strong extratropical cyclones in the Northern Hemisphere. This is following on from a number of studies analysing the TIL in idealised model simulations of baroclinic lifecycles. In both the case studies and the composites the authors find that the TIL strength (i.e. the highest values of static stability above the tropopause) can be found in the region of low isentropic PV that is advected cyclonically around the cyclone at upper levels. This is above the location of the ascent in the cyclones, where the clouds are identified. These results seem to be consistent with the previous studies. I have a few comments, queries and suggestions that the authors should consider."*

**Authors response:**

We thank the reviewer for the careful reading and interest in the study which helped to improve the paper. We reply to every comment in the order as they are listed in the review. We will respond in the following structure: First each comment from the reviewer, followed by our response, and finally the changes in the manuscript. The text coloured in cyan are the changes associated with this review, blue coloured text refers to the second review. The pages and lines indicate the position of the changes in the updated manuscript (the version without color-marked changes).
* * *
**Comment 1:** I found the introduction rather hard to read, with a lot of jargon that would be impenetrable to anyone new to the topic. I recommend reworking a lot of the language in the introduction section to make it clearer. For example, the 2nd paragraph on page 2 could be reworded.

**Reply to comment 1:** We agree that the introduction was written through the glasses of someone being rather familiar with the history of research on the topic of the tropopause inversion layer. We rephrased several formulations, reduced the technical description of numerical model experiment settings, and added descriptive caption-like sentences in between the paragraph to make the topic more approachable to someone who is less familiar with the phenomenon of the tropopause inversion layer. We however believe that some of the technical jargon is unavoidable when describing the approach and major outcome of the theoretical modelling studies on the TIL formation within baroclinic waves.

**Changes in the manuscript:**

**Page 2 Line 7:** This study focusses on the evolution of the TIL at midlatitudes, where the flow in the UTLS is largely dominated by baroclinic planetary and synoptic scale waves.  These waves play a major role concerning the formation and maintenance of the TIL in midlatitudes, and they have been subject of a variety of modelling studies on the TIL (Wirth, 2003, 2004; Wirth and Szabo, 2007; Erler and Wirth, 2011; Kunkel et al., 2014, 2016). ~~Idealised modelling studies showed that the TIL can be formed due to conservative dynamics. Wirth (2003, 2004) performed potential vorticity (PV) inversions on axisymmetric PV anomalies of different sign in an idealised background atmosphere, pointing out an adiabatic sharpening mechanism of the lower stratospheric temperature profile related to the convergence of the secondary circulation vertical wind in anticyclonic flow. They were furthermore able to show that the advection of enhanced static stability from low to high latitudes plays an important role for the lower stratospheric $N^2$ maximum in anticyclonic flow.Wirth and Szabo (2007) performed baroclinic life cycle simulations with a comprehensive numerical weather prediction model and were able to confirm the concept of an adiabatic sharpening mechanism of the tropopause.Following up on these results, Erler and Wirth (2011) performed adiabatic baroclinic life cycle simulations with the same setup and concluded that breaking of baroclinic waves is an important process for the irreversible and permanent formation of a residual TIL as evident in the zonal or temporal mean states.~~ The nonlinear interactions during the breaking of synoptic scale waves are

[revised manuscript text omitted]

. Recently, Flaounas et al. (2015) studied a set of 200 intensive Mediterranean cyclones based on a 20 year Weather Research and Forecasting (WRF) regional model data simulation with one focus among others on the UTLS PV forcing on the overall life cycle evolution and its synergy with the tropospheric development of the cyclones. To our knowledge the presented study is the first to focus on the TIL and associated features in the context of cyclone composites.
* * *
**Comment 2:** Also regarding the introduction, it could be made clearer exactly where the gaps lie that this paper seeks to address. At the bottom of page 1, "evidence for this relation still missing" implies that this paper will provide evidence, but I am unsure if this is a goal of the study. Some rewording might make the introduction clearer overall.

**Reply to comment 2:** After further consideration motivated by your comment we rephrased the paragraph in question to mitigate the implication on what the reader should expect from our analysis of the tropopause inversion layer. This change in combination with the changes made in the introduction section in reference to your first comment should clarify the major goals of this study.

15 We want to motivate the investigation of the role of the TIL concerning troposphere-stratosphere exchange processes. The TIL is commonly regarded as a transport barrier, in agreement with the fluiddynamical ramifications of a layer of enhanced static stability. However, one key result of our study is the potential for cross-tropopause exchange in the late stage of a baroclinic life cycle. Our analysis for the special case of the regions of enhanced static stability in late stage extratropical baroclinic waves shows a superimposed tendency towards dynamic instability due to large vertical wind shear in these regions.

20

**Changes in the manuscript:**

**Page 1 Line 23:** This co-location might imply a possible controlling function of the TIL for mixing in the UTLS, also based on the fluiddynamical ramifications of a layer of enhanced static stability, as large values of $N^2$ suppress vertical motion. Moreover, the TIL is essential for the vertical propagation of waves on different scales, ranging from small scale gravity waves to large scale Rossby waves (e.g., Birner, 2006; Sjoberg and Birner, 2014; Gisinger et al., 2017). The sharp jump in static stability at the tropopause from mean tropospheric values of $N^2 = 1 \times 10^{-4}\,\mathrm{s}^{-2}$ to mean stratospheric values of $N^2 = 4 \times 10^{-4}\,\mathrm{s}^{-2}$ or to the even larger values defining the TIL results in a maximum of the so called refractive index controlling the upward propagation of waves, and leading to partial or even total wave reflection at the tropopause. The overall role of the TIL concerning mixing processes between troposphere and stratosphere however is still not finally understood.

**Page 3 Line 5:** Furthermore, we investigate the mean flow features within regions of enhanced static stability with a focus on the role of the TIL for cross-tropopause exchange, including a physical mechanism potentially leading to dynamical insta-
35 bilities above the tropopause.
* * *
**Comment 3:** Page 2 line 19: The "residual TIL" is mentioned more than once. Is it possible to define what this looks like?

40 **Reply to comment 3:** We reconsidered the formulation and changed the wording from "residual TIL" to "background TIL", which is more fitting and also more self-explanatory. The changes concern several sentences in the introduction section. The updated introduction with marked changes is already attached to our reply to your first comment.
* * *
45 **Comment 4:** Page 2 line 32: This Kedzierski et al reference is never returned to in the discussion section, although I am sure the results from the present paper confirm those results (the TIL strongest within the ridges).

**Reply to comment 4:** We describe the work by Pilch Kedzierski et al. (2015) in our introduction section because it gives a demonstrative measurement based description of the TIL in mid-latitudes. The study analyses hemispheric snapshots of the

TIL distribution, as well as a statistical correlation analysis between the static stability above the tropopause and the upper tropospheric relative vorticity. We agree that the results from Pilch Kedzierski et al. (2015) match with our results concerning the synoptic scale variability of the TIL in mid-latitudes in ridges and troughs, but our analysis is specifically targeting late stage breaking baroclinic waves. Furthermore, we look at the shaping of the TIL in more detail than it would be possible from the global coverage rate of about 2000 daily GPS-RO profiles. We perceive the study of Pilch Kedzierski et al. (2015) as part of the basis for our work, where we try to expand their results by looking at the TIL evolution within troughs and ridges in a spatially and temporally higher resolved data set and describing the evolution. We added a reference to Pilch Kedzierski et al. (2015) in section 3.1 where we first describe the correlation between relative vorticity and static stability in troughs and ridges in our data set.

**Changes in the manuscript:**
**Page 10 Line 1:** This correlation especially holds true inside the $15°$ radius area and at the point in time of maximum cyclone intensity. This agrees well with the anticipated role of balanced dynamics (Wirth, 2003, 2004) and idealised baroclinic life cycle simulations with focus on the evolution of the TIL (e.g., Erler and Wirth, 2011). It furthermore agrees with the measurement based study by Pilch Kedzierski et al. (2015) concerning the correlation between the relative vorticity and the static stability within troughs and ridges.
* * *
**Comment 5:** Page 3, line 33: L91 and L137 are not vertical resolutions, but the number of levels. Could you give more information about the actual vertical resolution here?

**Reply to comment 5:** We changed the formulation from vertical resolution to vertical level count. We furthermore added a description on the geometric vertical resolution in the UTLS, the region of interest in our study.

**Changes in the manuscript:**
**Page 3 Line 31:** We use six hourly available analysis fields during the given time period, with a grid spacing of $0.25°$ in the horizontal, a vertical level count of L91 for the years 2010 until 2012, and L137 for 2013 and 2014. The abbreviation L91/L137 describes the 91/137 vertical level of the IFS's native hybrid sigma coordinates ranging from the earth's surface up to 0.01 hPa atmospheric pressure, with corresponding level spacing of typically 300-400 m in the UTLS region.
* * *
**Comment 6.:** Page 6, line 16: Could you make clearer what is meant by a "lapse rate tropopause based vertical coordinate"? Also page 7, line 7: what is meant by "the absolute height coordinate is recovered by calculating the mean tropopause height at each horizontal location"?

**Reply to comment 6:** We replaced the expression "lapse rate tropopause based vertical coordinate" by a description of the method, and added a reference to another study (Birner, 2006) that describes the method in detail.
In reference to your second question, the composite of the cyclones is calculated in lapse rate tropopause based vertical coordinates. Each horizontal coordinate $(\Theta, \phi)$ of each cyclone within the new coordinate system still carries the information of the absolute tropopause height, therefore a mean tropopause height can be calculated and assigned at this location within the composite. Hereby the level zero in the lapse rate tropopause based coordinates is replaced by the mean tropopause height, and all other vertical levels present a certain distance from this mean tropopause height. We added this description to the manuscript.

**Changes in the manuscript:**
**Page 7 Line 11:** The original ECMWF IFS variables in latitude-longitude coordinates are  interpolated onto a column covered by the new coordinate system, with the vertical coordinate using the lapse rate based tropopause as reference altitude (Birner, 2006). The tropopause height is defined as zero with negative/positive height values below/above and a vertical grid spacing of $\Delta z = 100$ m.

**Page 7 Line 16:** The ensemble of tropopause based  columns of variables from each cyclone is then averaged to create a three-dimensional composite of the flow in the vicinity of the cyclones. Subsequently, the mean absolute height coordinate is recovered as follows.  Each horizontal coordinate ($\Theta$, $\phi$) of each individual cyclone within the new coordinate system still carries the information about the absolute tropopause height at that location, therefore a mean tropopause height can be calculated and assigned to that horizontal location within the composite.
* * *
**Comment 7:** Page 7, line 8: What is meant by "horizontal or quasi-horizontal variables"? Does this just mean the horizontal composites of particular variables?

**Reply to comment 7:** We rephrased the expression to "horizontal or quasi-horizontal fields". While the expression "horizontal fields" commonly refers to variables on a plane of uniform geometric height or sometimes uniform pressure level, we use the expression quasi-horizontal to describe two-dimensional fields that are distinctly not horizontally aligned as defined above, e.g. the TIL strength above the highly variable tropopause height (especially in the vicinity of cyclones). The expression quasi-horizontal has been used in the same way in other studies (e.g., Wirth, 2003; Wirth and Szabo, 2007).

**Changes in the manuscript:**
**Page 7 Line 22:** In the special case of composites of horizontal or quasi-horizontal fields like the potential vorticity on an isentropic surface or the TIL strength, we first calculate the fields for each cyclone and then afterwards the mean.
* * *
**Comment 8:** Page 8, line 11: Please do not start sentences with numerals.

**Reply to comment 8:** We wrote out the numeral.

**Changes in the manuscript:**
**Page 8 Line 11:** Twenty-four hours before the cyclone reaches maximum intensity the mean sea level pressure already falls below 975 hPa.
* * *
**Comment 9:** Figure 3: The dotted MSLP contour is hard to distinguish.

**Reply to comment 9:** We updated all contour plot showing the mean seal level pressure, as well as their description. The solid lines now depict mean sea level pressure isolines from 1013 hPa downwards in steps of 5 hPa, and the dashed lines still depict isolines of mean sea level pressure larger than 1013 hPa in 5 hPa steps.
* * *
**Comment 10:** Page 8, line 30: Is the maximum $N^2$ above the tropopause the best measure of TIL strength? It does seem to correlate well with the PV pattern, but would an average value give similar but smoother results? Have you tested this?

**Reply to comment 10:** The concern about which TIL strength definition is the most useful was also expressed by the other reviewer, and we gave a detailed explanation in our reply to that comment on why we chose the specific TIL-definition used in our study. We refer to the third comment in the first review and our answer to this comment. We prepared an additional set of plots of the TIL strength evolution for our case studies (LC1 and LC2, see Figures R1 and R2). These contour plots are based on a vertical average of $N^2$ as the TIL strength. They show a largely similar evolution of the TIL on a synoptic scale but a lot of fine scale variability is filtered due to the average. We prefer the maximum in $N^2$ as the TIL strength definition because we work with high resolution data and want to preserve the fine scale variability in our analysis. Furthermore, our definition of the TIL-strength is in accordance with previous studies (e.g., Erler and Wirth, 2011; Pilch Kedzierski et al., 2015).

The validity of the TIL-strength definition we used is further confirmed by Fig. 12 which we added to the manuscript, on one hand as a response to comment 5 of the first review and on the other hand because it provides a better illustration of the issue that we tried to display in Fig. 11. We plotted a set of vertical tropopause-based cross-section composites similar to Figure 9 for different rotation angles around the cyclone centre. While these composites are only based on the lapse rate tropopause definition and not on the definition of the TIL-strength, they do agree very well with the composites of the maximum in $N^2$ above the tropopause from the individual cyclones. A description of this issue was also added to the manuscript as a response to comment 5 of the first review.

[Figure]

**Figure R1.** Evolution of the $N^2$ average based TIL strength for the LC2 resembling baroclinic wave breaking event. The grey contour shows the average of static stability over the vertical extent of 3 km above the thermal tropopause. Figure 1a) shows the TIL strength 24 hours before maximum intensity of the surface cyclone, b) at maximum intensity and c) 24 hours later.

[Figure]

**Figure R2.** As in Figure R1 but for the LC1 resembling baroclinic wave breaking event.

**Comment 11:** Figure 4: The caption seems to have incomplete sentences.

**Reply to comment 11:** We rephrased the caption, also due to the additional plot as a reply to comment 4 of the first review.

5    **Changes in the manuscript:**
**Page 11 Figure 4 caption:** Vertical cross-section over the North Atlantic on 14.10.2014 at  06:00 UTC, (a) at 42° N and (b) at 60° N. The filled contour as well as the thin solid black contour lines show static stability $N^2$ in steps of $1 \times 10^{-4}$ s$^{-2}$, dashed black lines show isentropes. The bold solid black line indicates the lapse rate tropopause, the dotted red line shows the 2 pvu isoline of potential vorticity.
* * *
**Comment 12:** Page 12, lines 9-10: I?m unsure of what this is referring to. Can you point out where the occlusion and the jet are located?

15    **Reply to comment 12:** After further consideration motivated by your comment, we prefer not to use the term occlusion for the case study of the cut-off cyclone, because this term commonly used in synoptic meteorology describes the seclusion of warm air from the ground due to the cold front moving faster than the warm front. The cyclone in question is migrating at rather low latitudes, therefore we find potential fronts to be less easy identified, and the description of the region of strongest TIL enhancement using the term occlusion (if justified) would still be confusing. We reworded the description, and added the 20  200 hPa wind maximum in Figure 6d (similar to Figure 3 and 5) to show the location of the jet (or jet streak).

   **Changes in the manuscript:**
**Page 13 Line 5:**  The regions of high static stability above the tropopause are located 25  north of the cyclone centre inside the flow with negative relative vorticity, as well as inside the flow that turns anticyclonically north-west of the cyclone centre and towards the jet maximum (as indicated by the blue contour lines in Fig. 6d).
* * *
**Comment 13:** Page 12, lines 17-18: This sentence is unclear. Did those authors look at the same events?

30

**Reply to comment 13:** The authors of Erler and Wirth (2011) focused on the formation of the so called residual TIL during adiabatic baroclinic life cycle simulations, as is evident in their study for example in Hovmöller diagrams or averaged vertical profiles of $N^2$. They show snapshots of the evolution of the TIL strength as well (Figure 4), but they do not go into much detail on the connection with the surface cyclones. It is however possible to interpret the results of our study about when and where 35  a strong TIL forms into the contour snapshots of $N^2_{max}$ of their simulations, especially in combination with the information about the surface pressure given in their Figure 2. We agree that this note is very specific, but we want to keep it because we considered it notable in case a reader is further interested in a comparisson between our study and idealised baroclinic life cycle simluations.

40
* * *
**Comment 14:** Pages 15-16: With the discussion of the Richardson number, it would be useful to the reader to have an explanation of what the results really mean. On page 17, lines 7-9, it seems to imply that despite the low $Ri$, turbulent mixing may occur, whereas in the discussion section (p18, lines 32- 33) it seems to say that because of the low $Ri$ turbulent mixing occurs. Please could you clarify this. In relation to this, are you suggesting that the strong TIL actually enhances the mixing 45  across the tropopause? This would be in opposition to the Hegglin et al. (2009) and Gettelman and Wang (2015) references on page 1 of the introduction.

**Reply to comment 14:** We added a description of the Richardson number in Section 4 (Page 17 Line 4), after the description of the maximum wind shear north of the cyclone centre within the region of enhanced $N^2$. This gives the reader a better

introduction on how increased wind shear and co-located high static stability work towards a convectively stably stratified but dynamically unstable flow. We furthermore rephrased the first paragraph you quoted (*originally page 17, lines 7-9*) to resolve the contradiction you pointed out.

Your second question in comment 14 is closely related to your second comment in this review, and we already rephrased the paragraph in question motivated by the latter. The motivation for these changes is explained in our reply to comment 2. We do not want to imply that a strong TIL generally enhances mixing across the tropopause, because a layer of enhanced static stability can act as a transport barrier, on one hand because it inhibits vertical motion, and on the other hand (and related to the first point) because it acts as a maximum of the refractive index controlling the upward propagation of atmospheric waves. For the specific case of the regions of enhanced static stability in late stage breaking baroclinic waves however we do see a tendency towards dynamic instability induced by vertical wind shear. The Richardson number with low values as an indicator for turbulent instability is composed of both constituents, $Ri = N^2 \, S^{-2}$. When the wind shear component is of dominating order, this could result in a shallow (Whiteway et al., 2004) but systematic troposphere-stratosphere exchange process in this region, despite the enhanced static stability. This however is up to this point only hypothetical, because based on the analysis tools used in this study we can not estimate the efficiency of such an exchange process, therefore we only present the tendency towards maximum wind shear in regions of enhanced static stability without going into to much detail about possible consequences. We are currently working on a detailed analysis of a case study in reference to this topic, which is in preparation to be submitted to ACPD.

**Changes in the manuscript:**

**Page 17 Line 5:** Based on these results, we calculate the dimensionless gradient Richardson number $Ri$. It is defined as the ratio of static stability $N^2$ and the vertical shear of the horizontal wind $S^2$, $Ri = N^2 S^{-2}$. Enhanced values of $N^2$ describe a stably stratified flow with suppressed vertical motion, while enhanced vertical shear of the horizontal wind can result in dynamical shear instability. Linear wave theory predicts a critical Richardson number of $Ri_c = 0.25$, where dynamic instability can develop in stably stratified flow with $Ri < Ri_c$.

**Page 17 Line 16:** A second region of minimum mean Richardson numbers is located right above the tropopause near the cyclone centre, and extends into the regions of maximum static stability. In this region the vertical wind shear $S^2$ is the dominating factor that works towards dynamic instability.

**Page 17 Line 33:** The lowermost stratosphere north of the cyclone centre exhibits Richardson numbers of $Ri = 5 - 10$ which are still well above the critical value of $Ri_c = 0.25$ for turbulent flow. These Richardson numbers however present an average of a highly non-linear measure over 76 cyclones. For individual cyclones Richardson numbers often exhibit much lower values in this region, eventually below the critical value $Ri_c$ and thus are indicative for dynamic instability. Also we would like to remind the reader that these quantities are derived from model data with a vertical grid spacing of about $\Delta z \approx 300$ m. The two features of interest ($N^2_{max}$ and $S^2_{max}$) typically exhibit a vertical extent of 1-3 km, therefore the model output profiles of $N^2$ and $S^2$ in the lowermost stratosphere can be expected to be a to a certain degree smoothed representation of the real atmosphere. A possible misrepresentation of the ratio between the two gradient-based measures $N^2$ and $S^2$ opens the possibility for an even larger range of Richardson numbers in the real flow. In conclusion, there is the possibility that turbulence is present even in regions of enhanced static stability in the lower stratosphere which might affect cross tropopause transport in these regions.
* * *
**Comment 15:** Page 18, line 26: "strong tropospheric updrafts"... I wonder here if what you mean is the ascent associated with the warm conveyor belt (e.g. Madonna et al ), which is where you would expect to see the diabatic heating. In Figure 3 it looks like the strongest TIL is clearly associated with the WCB at each time in the lifecycle. After the maximum intensity, the WCB anticyclonic outflow region (to the northeast of the cyclone) is the location of the strongest TIL. I think you need to include reference the WCB and how the observed features relate to it and the associated outflow. Ref: Madonna, E., H. Wernli, H. Joos, and O. Martius (2014a), Warm conveyor belts in the ERA-Interim dataset (1979? 2010). Part I: Climatology

and potential vorticity evolution, J. Clim., 27, 3?26, doi:10.1175/JCLI-D- 12-00720.1.

**Reply to comment 14:** Thank you for pointing out this connection. While we did not use any tools like trajectory analysis to define and to investigate the warm conveyor belt, the apparent co-location of the WCB outflow and the regions of enhanced static stability is very notable, and has been pointed out before (Kunkel et al., 2016). We added a description of this issue throughout three different points in the manuscript; in the analysis Section of the LC2 case study, in the analysis of the vertical cross-section composite of the ice water content and the vertical wind, and in the discussion Section.

**Changes in the manuscript:**

**Page 10 Line 10:** The regions of strongest enhancement of static stability above the tropopause in the second time step depicted in Fig. 3 as well as inside the deformed ridge in the third time step are associated to the regions commonly affected by the warm conveyor belt Madonna et al. (2014). The connection between the warm conveyor belt outflow and the regions of enhanced static stability above the tropopause is also a feature of the baroclinic life cycle simulations by Kunkel et al. (2016).

**Page 16 Line 14:** The ice clouds within and above the strong tropospheric updraft north of the cyclone centre reaching up to the tropopause with potential temperatures over $\Theta = 320$ K are features of the region where typically the warm conveyor belt outflow occurs (e.g., Madonna et al., 2014).

**Page 21 Line 4:** The regions of largest TIL enhancement are located north and northeast of the cyclone centre  above regions influenced by strong tropospheric updrafts and clouds reaching up to the tropopause, where in this central stage of the baroclinic life cycle typically the warm conveyor belt outflow occurs (e.g., Madonna et al., 2014; Kunkel et al., 2016). The high reaching clouds  indicate the importance of moist dynamical and radiative processes during the formation of the TIL (e.g., Randel et al., 2007; Kunkel et al., 2016).

**References**

[revised manuscript text omitted]

---

## Author Response (ED1)

**Authors Response**

We thank Amanda Maycock and the two anonymous referees for the interest in our study and the careful reading which helped to improve the paper. We hope that our replies to the comments from the referees answer the issues in a satisfying way, and that the changes in the manuscript motivated by the comments improved the paper. This document is structured as follows: First our point-by-point response to the first review, followed by the response to the second review (beginning at page 8), each in the following order: 1.) comment from the reviewer, 2.) our response and 3.) the changes in the manuscript. Changes marked in blue refer to the first review, cyan coloured text refers to the second review. A marked up manuscript version is attached below the responses.

**Point-by-point response to the first review**

───────────────────

**Comment 1:** Cyclone tracking: P5, lines 6 to 11. I appreciated the detailed description of the cyclone tracking algorithm but I am concerned about the effect of the projection and interpolation. The final composited feature in static stability looks like a North-South dipole. I am wondering what the effect of the original grid and interpolation does to this feature. E.g., does it enhance the dipole?

**Reply to comment 1:** The Lambert projection is solely part of the tracking algorithm. Once the tracks are defined, all data like e.g. the quasi-horizontal fields of the maximum in static stability $N^2$ above the thermal tropopause are extracted from the original ECMWF latitude-longitude-grid. Therefore the Lambert projection has no effect on the analysis of the North-South dipole feature of the static stability.

The interpolation onto an area-preserving Lambert projection is done based on the work of Hanley and Caballero (2012) who chose this specific projection to counteract the bias in effective zonal resolution when searching for local minima in the mean sea level pressure field (or any field for that reason). The higher zonal resolution at higher latitudes (with reference to the geometric distance between grid points) could be regarded as an advantage, representing denser information, but at the same time it represents an inconsistency between high and low latitudes. The interpolation on to the area preserving Lambert projection with uniform geometric grid (with 28 km grid spacing, representing a form of lowpass-filter for the higher latitudes) counteracts that fact, achieving a more consistent cyclone track density over the whole area.

───────────────────

**Comment 2:** Composites of cyclones: P6, line 11. Do you mean cylindrical rather than spheric? The data you have is on an equidistant grid (Is this gridding only done for the cyclone tracking or is this gridded data used for the subsequent composites). I am confused about which data is being interpolated to the red pillar since the underlying grey grid in figure 2 is not equidistant.

**Reply to comment 2:** The basis for the interpolation is defined by the rotation of a spherical polar coordinate surface onto the centre of a tracked cyclone. The pillar of gridded data interpolated on the new rotated grid with the height coordinate dz relative to the tropopause could indeed be described as a cylindrical polar coordinate system. The underlaying grey grid in Figure 2 represents the original latitude-longitude grid as provided by the ECMWF with equidistant latitude and longitude grid spacing, but the geometric distance depends on the latitude (the aforementioned bias in zonal resolution). These original fields are interpolated onto the rotated new spherical coordinates with spherical cap of $15°$ (red area). This achieves comparable areas independent of where each cyclone centre is located. We reworded the specific paragraph describing the interpolation method as follows:

**Changes in the manuscript:**

**Page 7 Line 4:** We select a subset of the gridded data as provided by the ECMWF for each cyclone by rotating the pole of a spheric polar coordinate system ($\Theta$, $\phi$) onto the centre of the cyclone and interpolated the original data onto that new grid. The horizontal resolution of the new coordinate system is set to $0.25°$ and the spherical cap radius to $\Theta_{max} = 15°$ (Fig. 2). The radius is chosen such that all relevant features around the cyclone centre are covered. Figure 2 illustrates the interpolation of the data from the original latitude-longitude grid provided by the ECMWF (light grey area for the whole sphere, dark grey for the limited area used in this study) onto an arbitrarily placed new spherical polar coordinate system (orange area).
* * *
**Comment 3:** LC1 and LC2 case studies: Seeing the case studies is helpful in interpreting the composites. Care should be taken in discussing the static stability strength, particularly with the discontinuities that are seen as a result of the analysis in fig 3c and 5c. I would be interested in seeing the average of $N^2$ (as opposed to $N^2_{max}$) in the region 3 km above the tropopause. Side note: Add your definition of $N^2_{max}$ to the caption in Fig 3c.

**Reply to comment 3:** We agree that a detailed discussion of the sub-synoptic-scale or even sub-meso scale features should be treated with caution when using a certain TIL strength definition, due to the discontinuities described. We nevertheless choose the maximum in $N^2$ above the tropopause as the TIL-strength definition for this study, because we are working with fine grid spacing data, and we want to highlight the variability of the static stability on different scales on one hand and show the difficulties when working with a certain definition of the TIL-strength on the other hand. The average within a certain vertical extent above the tropopause does not only filter a lot of information by default, but in particular for the feature of $N^2$ above the tropopause, because we know from previous work that larger maxima in $N^2$ above the tropopause tend to have a smaller vertical extent. Therefore an average over a defined vertical range above the tropopause could have a particular smoothing effect concerning even the large scale variability. We plotted the average of $N^2$ within 3 km above the tropopause for our case studies for comparison (Figure R1 and R2). Importantly, the analysis using a 3 km-averaged static stability shows very similar locations of the maximum TIL-structures on a synoptic scale, with the largest values in the regions of cyclonic wrap-up around the cyclone centre of air masses originating from lower latitudes. We did not add these plots to the manuscript, but we added the description of an average-based TIL strength definition at the end of section 2.1, to clarify that we deliberately decided to use another definition. We furthermore added the definition of the TIL strength to the caption in Figure 3.

**Changes in the manuscript:**
**Page 4 Line 18:** We  use this specific TIL strength definition, since the high resolution data shows large variability in the UTLS region, with often several maxima evident above the tropopause. Therefore, we find this definition of the TIL strength to be preferable over e.g. the first maximum in static stability above a threshold ($4 \times 10^{-4}$ s$^{-2}$, e.g., Gettelman and Wang, 2015). Another way to describe the static stability above the tropopause is to calculate the mean value of $N^2$ over a predefined vertical extent relative to the tropopause (e.g., Kunkel et al., 2014). While such a TIL strength definition achieves very similar results on a synoptic scale to the ones we will describe in Sect. 3, it also filters a lot of small scale variability and therefore partly neutralises the advantage of analysing high resolution data.

**Page 9 Fig. 3 caption:** Evolution of a baroclinic wave breaking event as seen in ECMWF IFS analysis data. The middle row represents the 14th of October 2014 at 06:00 UTC (the point in time of maximum cyclone intensity), the upper and lower row show the situation 24 hours prior and past the maximum intensity. Column (a) shows the pressure at mean  sea level  (solid lines $p_{msl} \leq 1013$ hPa, dashed lines $p_{msl} > 1013$ hPa, in steps of 5 hPa), (b) the IPV on 330 K, (c) the maximum of static static stability $N^2_{max}$ within 3 km above the thermal tropopause as indicator for the TIL strength, and (d) the relative vorticity at lapse rate tropopause height. Blue lines in middle row show 40 and 50 m s$^{-1}$ horizontal windspeed isolines at 200 hPa. Dashed black line shows the path of migration of the tracked cyclone, with the position of the cyclone centre at the point in time of the meteorological field displayed marked by the black x. Black circles show the 15° radius used for the composites. Note the deformation of the circles due to the mercator projection.

[Figure]

**Figure R1.** Evolution of the $N^2$ average based TIL strength for the LC2 resembling baroclinic wave breaking event. The grey contour shows the average of static stability over the vertical extent of 3 km above the thermal tropopause. Figure 1a) shows the TIL strength 24 hours before maximum intensity of the surface cyclone, b) at maximum intensity and c) 24 hours later.

[Figure]

**Figure R2.** As in Figure R1 but for the LC1 resembling baroclinic wave breaking event.

**Comment 4:** In figure 4, it may be more helpful to show the cross section at some latitude north of the cyclone centre since this is the region where there is an enhancement in $N^2$.

**Reply to comment 4:** We chose this cross-section in particular because it shows the small-scale stratospheric intrusion at around $12°$ W which is also visible as a discontinuity in the $N^2_{max}$ contour plot in Figure 3 in the middle row. It furthermore covers the tropopause jump between $30°$ W and $40°$ W. Both characteristics are the ones described in the paragraph associated with Figure 4. We included an additional cross-section (Figure R3) north of the cyclone centre into the manuscript to furthermore illustrate the variability of $N^2$ above the tropopause in the region of interest.

**Changes in the manuscript:**
**Page 10 Line 25:** The vertical cross-section in Fig. 4b shows the TIL structures at $60°$ N on the same day and north of the cyclone centre. It illustrates that the regions of enhanced static stability exhibit less variability at this latitude. They are located above the ridge between $40°$ W and $10°$ W, with the wave-like horizontal pattern that is also visible in Fig. 3c.

[Figure]

**Figure R3.** Vertical crosssection over the North Atlantic on 14.10.2014 at 06:00 UTC at $60°$ N. The filled contour as well as the thin solid black contour lines show static stability $N^2$ in steps of $1 \times 10^{-4}\text{s}^{-2}$, dashed black lines show isentropes. The bold solid black line indicates the lapse rate tropopause, the dotted red line shows the 2 pvu isoline of potential vorticity.

**Comment 5:** Composite analysis: I have some concerns about the compositing the values of $N^2_{max}$ and artefacts that might arise as a result of this. Have you looked at a number of cyclones in your composite to make sure that the features in N?2 are indeed present in most of them?

**Reply to comment 5:** This is a very valid concern which we shared during the data evaluation, especially since we know about the sensitivity of the result on the TIL-strength definition. We did in fact look at a large number of cyclones independently. While there is a large variability in the evolution of the upper tropospheric / lower stratospheric wrap-up around the cyclone centre (caused by the variability of the background flow, the cyclone evolution and the wave activity in the UTLS, as well as the variety of mechanism which influence the static stability at tropopause altitude). Most of the cyclones and especially the strong ones exhibit a comparable evolution of the static stability above the tropopause, with the wrapped-up North-South-dipole as the synoptic to mesoscale shaping. We furthermore estimate the mentioned discontinuities in the TIL-strength due the reasons described in the last paragraph of subsection 3.1 to be small and infrequent enough to be negligible for the large scale composite analysis. We updated Figure 11 and added Fig 12 to the manuscript with a set of vertical tropopause-based

cross-section composites of $S^2$ and $N^2$, which further illustrate the issue in a more comprehensive way. The plots complement Figure 9 and Figure 10b for different rotation angles around the cyclone centre, and while all these cross-section composites are solely based on the lapse rate tropopause definition, they do agree very well with the composites derived from the quasi-horizontal fields of the individual cyclones that depend on the definition of the TIL-strength and the strength of the wind shear (maximum of $N^2$ (Fig. 7b) / $S^2$ (Fig. 12) within 3 km above the tropopause). We added a description of this comparison to the manuscript.

**Changes in the manuscript:**

[revised manuscript text omitted]

**Comment 6:** Richardson number analysis: I find this very interesting. Do the corresponding plots for the case studies in section 3 show very low values of $Ri$ above the tropopause (regions with $Ri < 1$)?

**Reply to comment 6:** The regions of enhanced static stability within the individual cyclones are mostly dominated by large vertical shear of the horizontal wind which results in low Richardson numbers despite the enhanced $N^2$, as is the case for the LC1 and LC2 case study (Figure R4). Several cyclones in our dataset exhibit regions with even very low Richardson numbers ($Ri < 1$) at tropopause altitudes, in anticyclonic upper tropospheric flow inside the ridge of the baroclinic waves and with partial overlap into the regions of enhanced static stability. These regions however appear as transient and localised features. A detailed analysis of such a case is subject to a corresponding study we are working on, which is in preparation to be submitted to ACPD.

[Figure]

**Figure R4.** Minimum Richardson numbers $Ri_{min}$ within 3 km above the thermal tropopause, at the point in time of maximum cyclone intensity a.) for the LC2 resembling baroclinic wave, and b.) for the LC1 resembling baroclinic wave.

**Comment 7:** P18 line 31. The Richardson numbers found are not low enough for turbulent flow. I would suggest not making such a strong statement in the conclusion.

**Reply to comment 7:** We rephrased the paragraph to make the statement less strong and to clarify that we do not expect the whole region to be dynamically unstable (because the Richardson numbers are still too large as you pointed out). We rather want to note that the general tendency towards high vertical wind shear in the region of interest (considering the variability of $N^2$ and $S^2$) points toward the possibility for localised turbulence.

**Changes in the manuscript:**
**Page 21 Line 11:** Mean Richardson numbers calculated for these flow conditions  reveal a region of local minima in $Ri$ right above the tropopause at around 5° north from the cyclone centre in the composite.  Taking into account the variability of $N^2$ and $S^2$ for individual cyclones, this result points toward the possibility of a co-location between enhanced static stability above the tropopause and

localised regions of turbulent mixing of tropospheric and stratospheric air masses (Kunkel et al., 2016).

**Comment 8:** The colour scale on some of the figures could be chosen to be slightly more intuitive. E.g., Fig 10 (c) At first glance, I thought the red values were bigger.

**Reply to comment 8:** We agree that this plot can be counterintuitive due to the inverted color scale, but we want to highlight the region with the tendency toward turbulent motion (i.e. low Richardson numbers) and therefore want to keep the inverted colors for this particular Figure. We added a note on this to the Figure caption. We furthermore changed the color scale to a lower maximum value to make the plot more easy to read.

**Changes in the manuscript:**
**Page 18 Fig. 10 caption:** Vertical tropopause-based averaged cross-section as in Fig. 9, but only north of the cyclone centre. Left panel: filled contours show mean vertical velocity, solid thin black line the $\omega = 0\,\mathrm{Pa\,s^{-1}}$ isoline. Grey dashed lines are isolines of the cloud ice water content, steps as in Fig. 9. Middle panel: filled contour and solid black contour lines show squared vertical shear of horizontal wind $S^2$. Right panel: modified Richardson number mean $Ri$, with an inverted color scale (red represents low values) to highlight the tendency toward turbulent motion. Blue solid bold line in all three panels shows the $N_{mean}^2 = 6 \times 10^{-4}\,\mathrm{s^{-2}}$ isoline, dashed black lines the isentropes.

**Other comments:** P2, line 31. Fix reference - P4 line 3 "the a more", line 25 "oder" - P6 line 6, "North Africa an the" - P9 caption a) "seal level" - P12 line 7 "extend" - P13 lines 17 and 18 "This region of strong with..." Meaning unclear. - P14 line 8 "stronger" to more pronounced - P 16 fig 10 caption "Vertical crosssection as in figure 8" should be figure 9. also "north the cyclone", "of" missing.

**Reply:** We fixed the mistakes in the manuscript.

**Point-by-point response to the second review**
* * *
**Comment 1:** I found the introduction rather hard to read, with a lot of jargon that would be impenetrable to anyone new to the topic. I recommend reworking a lot of the language in the introduction section to make it clearer. For example, the 2nd paragraph on page 2 could be reworded.

**Reply to comment 1:** We agree that the introduction was written through the glasses of someone being rather familiar with the history of research on the topic of the tropopause inversion layer. We rephrased several formulations, reduced the technical description of numerical model experiment settings, and added descriptive caption-like sentences in between the paragraph to make the topic more approachable to someone who is less familiar with the phenomenon of the tropopause inversion layer. We however believe that some of the technical jargon is unavoidable when describing the approach and major outcome of the theoretical modelling studies on the TIL formation within baroclinic waves.

**Changes in the manuscript:**

[revised manuscript text omitted]

* * *
**Comment 2:** Also regarding the introduction, it could be made clearer exactly where the gaps lie that this paper seeks to address. At the bottom of page 1, "evidence for this relation still missing" implies that this paper will provide evidence, but I am unsure if this is a goal of the study. Some rewording might make the introduction clearer overall.

**Reply to comment 2:** After further consideration motivated by your comment we rephrased the paragraph in question to mitigate the implication on what the reader should expect from our analysis of the tropopause inversion layer. This change in combination with the changes made in the introduction section in reference to your first comment should clarify the major goals of this study.

We want to motivate the investigation of the role of the TIL concerning troposphere-stratosphere exchange processes. The TIL

is commonly regarded as a transport barrier, in agreement with the fluiddynamical ramifications of a layer of enhanced static stability. However, one key result of our study is the potential for cross-tropopause exchange in the late stage of a baroclinic life cycle. Our analysis for the special case of the regions of enhanced static stability in late stage extratropical baroclinic waves shows a superimposed tendency towards dynamic instability due to large vertical wind shear in these regions.

**Changes in the manuscript:**
**Page 1 Line 23:**  This co-location might imply a possible controlling function of the TIL for mixing in the UTLS, also based on the fluiddynamical ramifications of a layer of enhanced static stability, as large values of $N^2$ suppress vertical motion. Moreover, the TIL is essential for the vertical propagation of waves on different scales, ranging from small scale gravity waves to large scale Rossby waves (e.g., Birner, 2006; Sjoberg and Birner, 2014; Gisinger et al., 2017). The sharp jump in static stability at the tropopause from mean tropospheric values of $N^2 = 1 \times 10^{-4}\,\mathrm{s}^{-2}$ to mean stratospheric values of $N^2 = 4 \times 10^{-4}\,\mathrm{s}^{-2}$ or to the even larger values defining the TIL results in a maximum of the so called refractive index controlling the upward propagation of waves, and leading to partial or even total wave reflection at the tropopause. The overall role of the TIL concerning mixing processes between troposphere and stratosphere however is still not  understood.

**Page 3 Line 5:** Furthermore, we investigate the mean flow features within regions of enhanced static stability with a focus on the role of the TIL for cross-tropopause exchange, including a physical mechanism potentially leading to dynamical instabilities above the tropopause.
* * *
**Comment 3:** Page 2 line 19: The "residual TIL" is mentioned more than once. Is it possible to define what this looks like?

**Reply to comment 3:** We reconsidered the formulation and changed the wording from "residual TIL" to "background TIL", which is more fitting and also more self-explanatory. The changes concern several sentences in the introduction section. The updated introduction with marked changes is already attached to our reply to your first comment.
* * *
**Comment 4:** Page 2 line 32: This Kedzierski et al reference is never returned to in the discussion section, although I am sure the results from the present paper confirm those results (the TIL strongest within the ridges).

**Reply to comment 4:** We describe the work by Pilch Kedzierski et al. (2015) in our introduction section because it gives a demonstrative measurement based description of the TIL in mid-latitudes. The study analyses hemispheric snapshots of the TIL distribution, as well as a statistical correlation analysis between the static stability above the tropopause and the upper tropospheric relative vorticity. We agree that the results from Pilch Kedzierski et al. (2015) match with our results concerning the synoptic scale variability of the TIL in mid-latitudes in ridges and troughs, but our analysis is specifically targeting late stage breaking baroclinic waves. Furthermore, we look at the shaping of the TIL in more detail than it would be possible from the global coverage rate of about 2000 daily GPS-RO profiles. We perceive the study of Pilch Kedzierski et al. (2015) as part of the basis for our work, where we try to expand their results by looking at the TIL evolution within troughs and ridges in a spatially and temporally higher resolved data set and describing the evolution. We added a reference to Pilch Kedzierski et al. (2015) in section 3.1 where we first describe the correlation between relative vorticity and static stability in troughs and ridges in our data set.

**Changes in the manuscript:**
**Page 10 Line 1:** This correlation especially holds true inside the $15°$ radius area and at the point in time of maximum cyclone intensity. This agrees well with the anticipated role of balanced dynamics (Wirth, 2003, 2004) and idealised baroclinic life cycle simulations with focus on the evolution of the TIL (e.g., Erler and Wirth, 2011). It furthermore agrees with the measurement based study by Pilch Kedzierski et al. (2015) concerning the correlation between the relative vorticity and the static

stability within troughs and ridges.
* * *
**Comment 5:** Page 3, line 33: L91 and L137 are not vertical resolutions, but the number of levels. Could you give more information about the actual vertical resolution here?

**Reply to comment 5:** We changed the formulation from vertical resolution to vertical level count. We furthermore added a description on the geometric vertical resolution in the UTLS, the region of interest in our study.

**Changes in the manuscript:**
**Page 3 Line 31:** We use six hourly available analysis fields during the given time period, with a grid spacing of 0.25°in the horizontal, a vertical level count of L91 for the years 2010 until 2012, and L137 for 2013 and 2014. The abbreviation L91/L137 describes the 91/137 vertical level of the IFS's native hybrid sigma coordinates ranging from the earth's surface up to 0.01 hPa atmospheric pressure, with corresponding spacing of typically 300-400 m in the UTLS region.
* * *
**Comment 6.:** Page 6, line 16: Could you make clearer what is meant by a "lapse rate tropopause based vertical coordinate"? Also page 7, line 7: what is meant by "the absolute height coordinate is recovered by calculating the mean tropopause height at each horizontal location"?

**Reply to comment 6:** We replaced the expression "lapse rate tropopause based vertical coordinate" by a description of the method, and added a reference to another study (Birner, 2006) that describes the method in detail.
In reference to your second question, the composite of the cyclones is calculated in lapse rate tropopause based vertical coordinates. Each horizontal coordinate ($\Theta$, $\phi$) of each cyclone within the new coordinate system still carries the information of the absolute tropopause height, therefore a mean tropopause height can be calculated and assigned at this location within the composite. Hereby the level zero in the lapse rate tropopause based coordinates is replaced by the mean tropopause height, and all other vertical levels present a certain distance from this mean tropopause height. We added this description to the manuscript.

**Changes in the manuscript:**
**Page 7 Line 11:** The original ECMWF IFS variables in latitude-longitude coordinates are  interpolated onto a column covered by the new coordinate system, with the vertical coordinate using the lapse rate based tropopause as reference altitude (Birner, 2006). The tropopause height is defined as zero with negative/positive height values below/above and a vertical grid spacing of $\Delta z = 100$ m.

**Page 7 Line 16:** The ensemble of tropopause based  columns of variables from each cyclone is then averaged to create a three-dimensional composite of the flow in the vicinity of the cyclones. Subsequently, the mean absolute height coordinate is recovered as follows.  Each horizontal coordinate ($\Theta$, $\phi$) of each individual cyclone within the new coordinate system still carries the information about the absolute tropopause height at that location, therefore a mean tropopause height can be calculated and assigned to that horizontal location within the composite.
* * *
**Comment 7:** Page 7, line 8: What is meant by "horizontal or quasi-horizontal variables"? Does this just mean the horizontal composites of particular variables?

**Reply to comment 7:** We rephrased the expression to "horizontal or quasi-horizontal fields". While the expression "horizontal fields" commonly refers to variables on a plane of uniform geometric height or sometimes uniform pressure level, we use the expression quasi-horizontal to describe two-dimensional fields that are distinctly not horizontally aligned as defined above, e.g. the TIL strength above the highly variable tropopause height (especially in the vicinity of cyclones). The expression

quasi-horizontal has been used in the same way in other studies (e.g., Wirth, 2003; Wirth and Szabo, 2007).

**Changes in the manuscript:**
**Page 7 Line 22:** In the special case of composites of horizontal or quasi-horizontal fields like the potential vorticity on an isentropic surface or the TIL strength, we first calculate the fields for each cyclone and then afterwards the mean.
* * *
**Comment 8:** Page 8, line 11: Please do not start sentences with numerals.

**Reply to comment 8:** We wrote out the numeral.

**Changes in the manuscript:**
**Page 8 Line 11:** Twenty-four hours before the cyclone reaches maximum intensity the mean sea level pressure already falls below 975 hPa.
* * *
**Comment 9:** Figure 3: The dotted MSLP contour is hard to distinguish.

**Reply to comment 9:** We updated all contour plot showing the mean seal level pressure, as well as their description. The solid lines now depict mean sea level pressure isolines from 1013 hPa downwards in steps of 5 hPa, and the dashed lines still depict isolines of mean sea level pressure larger than 1013 hPa in 5 hPa steps.
* * *
**Comment 10:** Page 8, line 30: Is the maximum $N^2$ above the tropopause the best measure of TIL strength? It does seem to correlate well with the PV pattern, but would an average value give similar but smoother results? Have you tested this?

**Reply to comment 10:** The concern about which TIL strength definition is the most useful was also expressed by the other reviewer, and we gave a detailed explanation in our reply to that comment on why we chose the specific TIL-definition used in our study. We refer to the third comment in the first review and our answer to this comment. We prepared an additional set of plots of the TIL strength evolution for our case studies (LC1 and LC2, see Figures R1 and R2). These contour plots are based on a vertical average of $N^2$ as the TIL strength. They show a largely similar evolution of the TIL on a synoptic scale but a lot of fine scale variability is filtered due to the average. We prefer the maximum in $N^2$ as the TIL strength definition because we work with high resolution data and want to preserve the fine scale variability in our analysis. Furthermore, our definition of the TIL-strength is in accordance with previous studies (e.g., Erler and Wirth, 2011; Pilch Kedzierski et al., 2015). The validity of the TIL-strength definition we used is further confirmed by Fig. 12 which we added to the manuscript, on one hand as a response to comment 5 of the first review and on the other hand because it provides a better illustration of the issue that we tried to display in Fig. 11. We plotted a set of vertical tropopause-based cross-section composites similar to Figure 9 for different rotation angles around the cyclone centre. While these composites are only based on the lapse rate tropopause definition and not on the definition of the TIL-strength, they do agree very well with the composites of the maximum in $N^2$ above the tropopause from the individual cyclones. A description of this issue was also added to the manuscript as a response to comment 5 of the first review.
* * *
**Comment 11:** Figure 4: The caption seems to have incomplete sentences.

**Reply to comment 11:** We rephrased the caption, also due to the additional plot as a reply to comment 4 of the first review.

**Changes in the manuscript:**
**Page 11 Figure 4 caption:** Vertical cross-section over the North Atlantic on 14.10.2014 at  06:00 UTC, (a)

at 42° N and (b) at 60° N. The filled contour as well as the thin solid black contour lines show static stability $N^2$ in steps of $1 \times 10^{-4}\,\text{s}^{-2}$, dashed black lines show isentropes. The bold solid black line indicates the lapse rate tropopause, the dotted red line shows the 2 pvu isoline of potential vorticity.
* * *
**Comment 12:** Page 12, lines 9-10: I?m unsure of what this is referring to. Can you point out where the occlusion and the jet are located?

**Reply to comment 12:** After further consideration motivated by your comment, we prefer not to use the term occlusion for the case study of the cut-off cyclone, because this term commonly used in synoptic meteorology describes the seclusion of warm air from the ground due to the cold front moving faster than the warm front. The cyclone in question is migrating at rather low latitudes, therefore we find potential fronts to be less easy identified, and the description of the region of strongest TIL enhancement using the term occlusion (if justified) would still be confusing. We reworded the description, and added the 200 hPa wind maximum in Figure 6d (similar to Figure 3 and 5) to show the location of the jet (or jet streak).

**Changes in the manuscript:**
**Page 13 Line 5:**  The regions of high static stability above the tropopause are located north of the cyclone centre inside the flow with negative relative vorticity, as well as inside the flow that turns anticyclonically north-west of the cyclone centre and towards the jet maximum (as indicated by the blue contour lines in Fig. 6d).
* * *
**Comment 13:** Page 12, lines 17-18: This sentence is unclear. Did those authors look at the same events?

**Reply to comment 13:** The authors of Erler and Wirth (2011) focused on the formation of the so called residual TIL during adiabatic baroclinic life cycle simulations, as is evident in their study for example in Hovmöller diagrams or averaged vertical profiles of $N^2$. They show snapshots of the evolution of the TIL strength as well (Figure 4), but they do not go into much detail on the connection with the surface cyclones. It is however possible to interpret the results of our study about when and where a strong TIL forms into the contour snapshots of $N^2_{max}$ of their simulations, especially in combination with the information about the surface pressure given in their Figure 2. We agree that this note is very specific, but we want to keep it because we considered it notable in case a reader is further interested in a comparisson between our study and idealised baroclinic life cycle simluations.
* * *
**Comment 14:** Pages 15-16: With the discussion of the Richardson number, it would be useful to the reader to have an explanation of what the results really mean. On page 17, lines 7-9, it seems to imply that despite the low $Ri$, turbulent mixing may occur, whereas in the discussion section (p18, lines 32- 33) it seems to say that because of the low $Ri$ turbulent mixing occurs. Please could you clarify this. In relation to this, are you suggesting that the strong TIL actually enhances the mixing across the tropopause? This would be in opposition to the Hegglin et al. (2009) and Gettelman and Wang (2015) references on page 1 of the introduction.

**Reply to comment 14:** We added a description of the Richardson number in Section 4, after the description of the maximum wind shear north of the cyclone centre within the region of enhanced $N^2$. This gives the reader a better introduction on how increased wind shear and co-located high static stability work towards a convectively stably stratified but dynamically unstable flow. We furthermore rephrased the first paragraph you quoted to resolve the contradiction you pointed out.
Your second question in comment 14 is closely related to your second comment in this review, and we already rephrased the paragraph in question motivated by the latter. The motivation for these changes is explained in our reply to comment 2. We do not want to imply that a strong TIL generally enhances mixing across the tropopause, because a layer of enhanced static stability can act as a transport barrier, on one hand because it inhibits vertical motion, and on the other hand (and related to the first

point) because it acts as a maximum of the refractive index controlling the upward propagation of atmospheric waves. For the specific case of the regions of enhanced static stability in late stage breaking baroclinic waves however we do see a tendency towards dynamic instability induced by vertical wind shear. The Richardson number with low values as an indicator for turbulent

5   instability is composed of both constituents, $Ri = N^2\,S^{-2}$. When the wind shear component is of dominating order, this could result in a shallow (Whiteway et al., 2004) but systematic troposphere-stratosphere exchange process in this region, despite the enhanced static stability. This however is up to this point only hypothetical, because based on the analysis tools used in this study we can not estimate the efficiency of such an exchange process, therefore we only present the tendency towards maximum wind shear in regions of enhanced static stability without going into to much detail about possible consequences. We are cur-

10  rently working on a detailed analysis of a case study in reference to this topic, which is in preparation to be submitted to ACPD.

**Changes in the manuscript:**
**Page 17 Line 5:** Based on these results, we calculate the dimensionless gradient Richardson number $Ri$. It is defined as the ratio of static stability $N^2$ and the vertical shear of the horizontal wind $S^2$, $Ri = N^2S^{-2}$. Enhanced values of $N^2$ describe a

15  stably stratified flow with suppressed vertical motion, while enhanced vertical shear of the horizontal wind can result in dynamical shear instability. Linear wave theory predicts a critical Richardson number of $Ri_c = 0.25$, where dynamic instability can develop in stably stratified flow with $Ri < Ri_c$.

**Page 17 Line 16:** A second region of minimum mean Richardson numbers is located right above the tropopause near the

20  cyclone centre, and extends into the regions of maximum static stability. In this region the vertical wind shear $S^2$ is the dominating factor that works towards dynamic instability.

**Page 17 Line 33:**

25   The lowermost stratosphere north of the cyclone centre exhibits Richardson numbers of $Ri = 5 - 10$ which are still well above the critical value of $Ri_c = 0.25$ for turbulent flow. These Richardson numbers however present an average of a highly non-linear measure over 76 cyclones. For individual cyclones Richardson numbers often exhibit much lower values in this region,  below the critical value $Ri_c$ and thus are indicative  dynamic instability. Also we would like to remind the reader that these quantities are derived from model data with a vertical grid spacing of about $\Delta z \approx 300$ m. The two features

30  of interest ($N^2_{max}$ and $S^2_{max}$) typically exhibit a vertical extent of 1-3 km, therefore the model output profiles of $N^2$ and $S^2$ in the lowermost stratosphere can be expected to be a to a certain degree smoothed representation of the real atmosphere. A possible misrepresentation of the ratio between the two gradient-based measures $N^2$ and $S^2$ opens the possibility for an even larger range of Richardson numbers in the real flow.  In conclusion, there is the possibility that turbulence is present even in regions of enhanced static stability in the lower stratosphere which might affect cross tropopause transport in these

35  regions.
* * *
**Comment 15:** Page 18, line 26: "strong tropospheric updrafts"... I wonder here if what you mean is the ascent associated with the warm conveyor belt (e.g. Madonna et al ), which is where you would expect to see the diabatic heating. In Figure 3

40  it looks like the strongest TIL is clearly associated with the WCB at each time in the lifecycle. After the maximum intensity, the WCB anticyclonic outflow region (to the northeast of the cyclone) is the location of the strongest TIL. I think you need to include reference the WCB and how the observed features relate to it and the associated outflow. Ref: Madonna, E., H. Wernli, H. Joos, and O. Martius (2014a), Warm conveyor belts in the ERA-Interim dataset (1979? 2010). Part I: Climatology and potential vorticity evolution, J. Clim., 27, 3?26, doi:10.1175/JCLI-D- 12-00720.1.

45

**Reply to comment 14:** Thank you for pointing out this connection. While we did not use any tools like trajectory analysis to define and to investigate the warm conveyor belt, the apparent co-location of the WCB outflow and the regions of enhanced static stability is very notable, and has been pointed out before (Kunkel et al., 2016). We added a description of this issue throughout three different points in the manuscript; in the analysis Section of the LC2 case study, in the analysis of the vertical

cross-section composite of the ice water content and the vertical wind, and in the discussion Section.

**Changes in the manuscript:**

**Page 10 Line 10:** The regions of strongest enhancement of static stability above the tropopause in the second time step depicted in Fig. 3 as well as inside the deformed ridge in the third time step are associated to the regions commonly affected by the warm conveyor belt Madonna et al. (2014). The connection between the warm conveyor belt outflow and the regions of enhanced static stability above the tropopause is also a feature of the baroclinic life cycle simulations by Kunkel et al. (2016).

**Page 16 Line 14:** The ice clouds within and above the strong tropospheric updraft north of the cyclone centre reaching up to the tropopause with potential temperatures over $\Theta = 320$ K are features of the region where typically the warm conveyor belt outflow occurs (e.g., Madonna et al., 2014).

**Page 21 Line 4:** The regions of largest TIL enhancement are located north and northeast of the cyclone centre  above regions influenced by strong tropospheric updrafts and clouds reaching up to the tropopause, where in this central stage of the baroclinic life cycle typically the warm conveyor belt outflow occurs (e.g., Madonna et al., 2014; Kunkel et al., 2016). The high  clouds   importance of moist dynamical and radiative processes during the formation of the TIL (e.g., Randel et al., 2007; Kunkel et al., 2016).

**References**

[revised manuscript text omitted]

---

## Author Response (AR2)

**Authors Response**

We thank Amanda Maycock for the thorough read of our study which helped to finalise the paper. We corrected the minor typos and grammatical corrections as suggested, and attached a marked up manuscript version below.

[revised manuscript text omitted]